# Contextual Bandits for Unbounded Context Distributions

**Puning Zhao** [1 2]  **Rongfei Fan** [3]  **Shaowei Wang** [4]  **Li Shen** [1 2]  **Qixin Zhang** [5]  **Zong Ke** [6]  **Tianhang Zheng** [7]

## Abstract

Nonparametric contextual bandit is an important model of sequential decision making problems. Under $\alpha$-Tsybakov margin condition, existing research has established a regret bound of $\tilde{O}\left(T^{1-\frac{\alpha+1}{d+2}}\right)$ for bounded supports. However, the optimal regret with unbounded contexts has not been analyzed. The challenge of solving contextual bandit problems with unbounded support is to achieve both exploration-exploitation tradeoff and bias-variance tradeoff simultaneously. In this paper, we solve the nonparametric contextual bandit problem with unbounded contexts. We propose two nearest neighbor methods combined with UCB exploration. The first method uses a fixed $k$. Our analysis shows that this method achieves minimax optimal regret under a weak margin condition and relatively light-tailed context distributions. The second method uses adaptive $k$. By a proper data-driven selection of $k$, this method achieves an expected regret of $\tilde{O}\left(T^{1-\frac{(\alpha+1)\beta}{\alpha+(d+2)\beta}} + T^{1-\beta}\right)$, in which $\beta$ is a parameter describing the tail strength. This bound matches the minimax lower bound up to logarithm factors, indicating that the second method is approximately optimal.

## 1. Introduction

Multi-armed bandit (Robbins, 1952; Lai & Robbins, 1985) is an important sequential decision problem that has been extensively studied (Agrawal, 1995; Auer et al., 2002; Garivier & Cappé, 2011). In many practical applications such as recommender systems and information retrieval in healthcare

and finance (Bouneffouf et al., 2020), decision problems are usually modeled as contextual bandits (Woodroofe, 1979), in which the reward depends on some side information, called contexts. At the $t$-th iteration, the decision maker observes the context $\mathbf{X}_t$, and then pulls an arm $A_t \in \mathcal{A}$ based on $\mathbf{X}_t$ and the previous trajectory $(\mathbf{X}_i, A_i), i = 1, \ldots, t-1$. Many research assume linear rewards (Abbasi-Yadkori et al., 2011; Bastani & Bayati, 2020; Bastani et al., 2021; Qian et al., 2023; Langford & Zhang, 2007; Dudik et al., 2011; Chu et al., 2011; Li et al., 2010), which is restrictive and may not fit well into practical scenarios. Consequently, in recent years, nonparametric contextual bandits have received significant attention, which does not make any parametric assumption about the reward functions (Perchet & Rigollet, 2013; Guan & Jiang, 2018; Gur et al., 2022; Blanchard et al., 2023; Suk & Kpotufe, 2023; Suk, 2024; Cai et al., 2024).

Despite significant progress on nonparametric contextual bandits, existing studies focus only on the case with bounded contexts, and the probability density functions (pdf) of the contexts are required to be bounded away from zero. However, many practical applications often involve unbounded contexts, such as healthcare (Durand et al., 2018), dynamic pricing (Misra et al., 2019) and recommender systems (Zhou et al., 2017). In particular, the contexts may follow a heavy-tailed distribution (Zangerle & Bauer, 2022), which is significantly different from bounded contexts. Therefore, to bridge the gap between theoretical studies and practical applications of contextual bandits, an in-depth theoretical study of unbounded contexts is crucially needed. Compared with bounded contexts, heavy-tailed context distribution requires the learning method to be adaptive to the pdf of contexts, in order to balance the bias and variance of the estimation of reward functions. On the other hand, compared with existing works on nonparametric classification and regression with identically and independently distributed (i.i.d) data, bandit problems require us to achieve a good balance between exploration and exploitation, thus the learning method needs to be adaptive to the suboptimality gap of reward functions. Therefore, the main challenge of solving nonparametric contextual bandit problems with unbounded contexts is to achieve both bias-variance tradeoff and exploration-exploitation tradeoff using a single algorithm.

In this paper, we solve the nonparametric contextual ban-

---

[1]Shenzhen Campus of Sun Yat-sen University, Shenzhen, China [2]Guangming Laboratory, Shenzhen, China [3]Beijing Institute of Technology, Beijing, China [4]Guangzhou University, Guangzhou, China [5]Nanyang Technological University, Singapore [6]National University of Singapore, Singapore [7]Zhejiang University, Hangzhou, China. Correspondence to: Li Shen <shenli6@mail.sysu.edu.cn>.

*Proceedings of the $42^{nd}$ International Conference on Machine Learning*, Vancouver, Canada. PMLR 267, 2025. Copyright 2025 by the author(s).

| | Method | Bound of expected regret | |
|---|---|---|---|
| | | Bounded context | Heavy-tailed context |
| (Rigollet & Zeevi, 2010) | UCBogram | $\tilde{O}\left(T^{1-\min\left\{\frac{\alpha+1}{d+2},\frac{2}{d+2}\right\}}\right)$ | None |
| (Perchet & Rigollet, 2013) | ABSE | $\tilde{O}\left(T^{1-\frac{\alpha+1}{d+2}}\right)$ | None |
| (Gur et al., 2022) | SACB | $\tilde{O}\left(T^{1-\frac{\alpha+1}{d+2}}\right)$ | None |
| (Guan & Jiang, 2018) | kNN-UCB | $\tilde{O}\left(T^{\frac{d+1}{d+2}}\right)$ | None |
| (Reeve et al., 2018) | kNN-UCB | $\tilde{O}\left(T^{1-\frac{\alpha+1}{d+2}}\right)$ | None |
| This work | kNN-UCB[1] | $\tilde{O}\left(T^{\max\left\{1-\frac{\alpha+1}{d+2},\frac{2}{\alpha+3}\right\}}\right)$ | $\tilde{O}\left(T^{1-\frac{\beta\min(d,\alpha+1)}{\min(d-1,\alpha)+\max(1,d\beta)+2\beta}}\right)$ |
| This work | Adaptive kNN-UCB | $\tilde{O}\left(T^{1-\frac{\alpha+1}{d+2}}\right)$ | $\tilde{O}\left(T^{1-\min\left\{\frac{(\alpha+1)\beta}{\alpha+(d+2)\beta},\beta\right\}}\right)$ |
| Minimax lower bound | | $\Omega\left(T^{1-\frac{\alpha+1}{d+2}}\right)$ | $\Omega\left(T^{1-\min\left\{\frac{(\alpha+1)\beta}{\alpha+(d+2)\beta},\beta\right\}}\right)$ |

*Table 1.* Comparison of the $T$-step expected cumulative regrets of learning algorithms for nonparametric contextual bandits with Lipschitz reward function under $\alpha$-Tsybakov margin condition and tail parameter $\beta$ (Assumption 1(a) and (b)).

dit problem with heavy-tailed contexts. To begin with, we derive a minimax lower bound that characterizes the theoretical limits of contextual bandit learning. We then propose a relatively simple method that uses fixed $k$ combined with upper confidence bound (UCB) exploration and derive the bound of expected regret. Even for bounded contexts, our method improves over an existing nearest neighbor method (Guan & Jiang, 2018) for large margin parameter $\alpha$, since our method uses an improved UCB calculation which is more adaptive to the suboptimality gap of reward functions. Despite such progress, there is still some gap between the regret bound and the minimax lower bound, indicating room for further improvement. To close such a gap, we further propose a new adaptive nearest neighbor approach, which selects $k$ adaptively based on the density of samples and the suboptimality gap of reward functions. Our analysis shows that the regret bound of this new method nearly matches the minimax lower bound up to logarithmic factors, indicating that this method is approximately minimax optimal.

The general guidelines of our adaptive kNN method are summarized as follows. Firstly, with higher context pdf, we use larger $k$, and vice versa. Secondly, given a specific context, if the value of an action is far away from optimal (i.e. large suboptimality gap), then we use smaller $k$, and vice versa. Such a choice of $k$ achieves a good tradeoff between estimation bias and variance. With a lower pdf or larger suboptimality gap, the samples are relatively more sparse. As a result, a large bias may happen due to large kNN distances. Therefore, we use smaller $k$ to control the bias. On the contrary, with a higher pdf or smaller suboptimality gap, the samples are dense and thus we can use larger $k$ to reduce the variance. Note that the pdf and the

suboptimality gap are unknown to the learner. Therefore, we design a method, such that the value of $k$ is selected by a data-driven manner, based on the density of existing samples.

### 1.1. Contribution

The contributions of this paper are summarized as follows.

- We derive the minimax lower bound of nonparametric contextual bandits with heavy-tailed context distributions.

- We propose a simple kNN method with UCB exploration. The regret bound matches the minimax lower bound with small $\alpha$ and large $\beta$.

- We propose an adaptive kNN method, such that $k$ is selected based on previous steps. The regret bound matches the minimax lower bound under all parameter regimes.

Our results and the comparison with related works are summarized in Table 1. In general, to the best of our knowledge, our work is the first attempt to handle heavy-tailed context distribution in contextual bandit problems. In particular, our new proposed adaptive kNN method achieves the minimax lower bound for the first time. The proofs of all theoretical results in the paper are shown in the supplementary material.

## 2. Related Work

In this section, we briefly review the related works about contextual bandits and nearest neighbor methods.

**Nonparametric contextual bandits with bounded contexts.** (Yang & Zhu, 2002) first introduced the nonparametric contextual bandit problem, proposed an $\epsilon$-greedy ap-

---

[1]Despite that the name kNN-UCB is the same as (Guan & Jiang, 2018) and (Reeve et al., 2018), the calculations of UCB are different between these methods. See details in the "Nearest neighbor method with fixed $k$" section.

proach and proved the consistency. (Rigollet & Zeevi, 2010) derived a minimax lower bound on the regret, and showed that this bound is achievable by a UCB method. (Perchet & Rigollet, 2013) proposed Adaptively Binned Successive Elimination (ABSE), which adapts to the unknown margin parameter. (Qian & Yang, 2016) proposed a kernel estimation method. (Hu et al., 2020) analyzed nonparametric bandit problem under general Hölder smoothness assumption. (Gur et al., 2022) proposed Smoothness-Adaptive Contextual Bandits (SACB), which is adaptive to the smoothness parameter. (Slivkins, 2014; Suk & Kpotufe, 2023; Akhavan et al., 2024; Ghosh et al., 2024; Komiyama et al., 2024; Suk, 2024) analyzed the problem of dynamic regret. Furthermore, (Wanigasekara & Yu, 2019; Locatelli & Carpentier, 2018; Krishnamurthy et al., 2020; Zhu et al., 2022) discussed the case with continuous actions.

**Nearest neighbor methods.** Nearest neighbor classification has been analyzed in (Chaudhuri & Dasgupta, 2014; Döring et al., 2018) for bounded support of features. (Gadat et al., 2016; Kpotufe, 2011; Cannings et al., 2020; Zhao & Lai, 2021b;a) proposed adaptive nearest neighbor methods for heavy-tailed feature distributions. (Guan & Jiang, 2018) proposed kNN-UCB method for contextual bandits and proved a regret bound $\tilde{O}(T^{\frac{1+d}{2+d}})$.

Compared with existing methods on nonparametric contextual bandits, for unbounded contexts, the methods need to adapt to different density levels of contexts and achieve a better bias and variance tradeoff in the estimation of reward functions. Moreover, existing works on nonparametric classification can not be easily extended here, since the samples are no longer i.i.d and we now need to bound the regret instead of the estimation error. These factors introduce new technical difficulties in theoretical analysis. In this work, to address these challenges, we design new algorithms that are adaptive to both the pdf and the suboptimality of reward functions and provide a corresponding theoretical analysis.

## 3. Preliminaries

Denote $\mathcal{X}$ as the space of contexts, and $\mathcal{A}$ as the space of actions. Throughout this paper, we discuss the case with infinite $\mathcal{X}$ and finite $\mathcal{A}$. At the $t$-th step, the context $\mathbf{X}_t$ is a random variable drawn from a distribution with probability density function (pdf) $f$. Then the agent takes action $A_t \in \mathcal{A}$ and receive reward $Y_t$:

$$Y_t = \eta_{A_t}(\mathbf{X}_t) + W_t, \tag{1}$$

in which $\eta_a(\mathbf{x})$ for $a \in \mathcal{A}$ and $\mathbf{x} \in \mathcal{X}$ is an unknown expected reward function, and $W_t$ denotes the noise, with $\mathbb{E}[W_t|\mathbf{X}_{1:t}, A_{1:t}] = 0$.

Throughout this paper, define

$$\eta^*(\mathbf{x}) = \max_a \eta_a(\mathbf{x}) \tag{2}$$

as the maximum expected reward of context $\mathbf{x}$. For any suboptimal action $a$, $\eta_a(\mathbf{x}) < \eta^*(\mathbf{x})$. Correspondingly, $\eta^*(\mathbf{x}) - \eta_a(\mathbf{x})$ is called suboptimality gap.

The performance of an algorithm is evaluated by the expected regret

$$R = \mathbb{E}\left[\sum_{t=1}^{T} (\eta^*(\mathbf{X}_t) - \eta_{A_t}(\mathbf{X}_t))\right]. \tag{3}$$

We then present the assumptions needed for the analysis. To begin with, we state some basic conditions in Assumption 1.

**Assumption 1.** *There exists some constants $C_\alpha$, $\sigma$, $L$, such that*

*(a) (Tsybakov margin condition) For some $\alpha \leq d$, for all $a \in \mathcal{A}$ and $u > 0$, $P(0 < \eta^*(\mathbf{X}) - \eta_a(\mathbf{X}) < u) \leq C_\alpha u^\alpha$;*

*(b) $W_t$ is subgaussian with parameter $\sigma^2$, i.e. $\mathbb{E}[e^{\lambda W_i}] \leq e^{\frac{1}{2}\lambda^2\sigma^2}$;*

*(c) For all $a$, $\eta_a$ is Lipschitz with constant $L$, i.e. for any $x$ and $\mathbf{x}'$, $|\eta_a(\mathbf{x}) - \eta_a(\mathbf{x}')| \leq L \|\mathbf{x} - \mathbf{x}'\|$.*

Now we comment on these assumptions. (a) is the Tsybakov margin condition, which was first introduced in (Audibert & Tsybakov, 2007) for classification problems, and then used in contextual bandit problems (Perchet & Rigollet, 2013). Note that $P(0 < \eta^*(\mathbf{X}) - \eta_a(\mathbf{X}) < t) \leq 1$ always hold, thus for any $\eta_a$, (a) holds with $C_\alpha = 1$ and $\alpha = 0$. Therefore, this assumption is nontrivial only if it holds with some $\alpha > 0$. Moreover, we only consider the case with $\alpha \leq d$ here. If $\alpha > d$, then an arm is either always or never optimal, thus it is easy to achieve logarithmic regret (see (Perchet & Rigollet, 2013), Proposition 3.1). An additional remark is that in (Perchet & Rigollet, 2013; Reeve et al., 2018), the margin assumption is $P(0 < \eta^*(\mathbf{X}) - \eta_s(\mathbf{X}) < t) \lesssim t^\alpha$, in which $\eta_s(\mathbf{x})$ is the second largest one among $\{\eta_a(\mathbf{x})|a \in \mathcal{A}\}$. Our assumption (a) is slightly weaker than existing ones since we only impose margin conditions on the suboptimality gap $\eta^*(\mathbf{x}) - \eta_a(\mathbf{x})$ for each $a$ separately, instead of on the minimum suboptimality gap. In (b), following existing works (Reeve et al., 2018), we assume that the noise has light tails. (c) is a common assumption for various literatures on nonparametric estimation (Mai & Johansson, 2021). It is possible to extend this work to a more general Hölder smoothness assumption by adaptive nearest neighbor weights (Cannings et al., 2020). In this paper, we focus only on Lipschitz continuity for convenience.

Assumption 2 is designed for the case that the contexts have bounded support.

**Assumption 2.** *$f(\mathbf{x}) \geq c$ for all $\mathbf{x} \in \mathcal{X}$, in which $f$ is the pdf of contexts.*

In Assumption 2, the pdf $f$ is required to be bounded away from zero, which is also made in (Perchet & Rigollet, 2013; Guan & Jiang, 2018; Reeve et al., 2018). Note that even for estimation with i.i.d data, this assumption is common (Audibert & Tsybakov, 2007; Döring et al., 2018; Gao et al., 2018).

We then show some assumptions for heavy-tailed distributions.

**Assumption 3.** *(a) For any $u > 0$, $P(f(\mathbf{X}) \leq u) \leq C_\beta u^\beta$ for some constants $C_\beta$ and $\beta$;*

*(b) The difference of regret function $\eta$ among all actions are bounded, i.e. $\sup_{\mathbf{x} \in \mathcal{X}}(\eta^*(\mathbf{x}) - \min_a \eta_a(\mathbf{x})) \leq M$ for some constant $M$.*

(a) is a common tail assumption for nonparametric statistics, which has been made in (Gadat et al., 2016; Zhao & Lai, 2021b). $\beta$ describes the tail strength. Smaller $\beta$ indicates that the context distribution has heavy tails, and vice versa. To further illustrate this assumption, we show several examples.

**Example 1.** *If $f$ has bounded support $\mathcal{X}$, then Assumption 3(a) holds with $C_\beta = V(\mathcal{X})$ and $\beta = 1$, in which $V(\mathcal{X})$ is the volume of the support set $\mathcal{X}$.*

**Example 2.** *If $f$ has $p$-th bounded moment, i.e. $\mathbb{E}[\|\mathbf{X}\|^p] < \infty$, then for all $\beta < p/(p+d)$, there exists a constant $C_\beta$ such that Assumption 3(a) holds. In particular, for subgaussian or subexponential random variables, Assumption 3(a) holds for all $\beta < 1$.*

*Proof.* The analysis of these examples and other related discussions are shown in the supplementary material. □

It worths mentioning that although the growth rate of the regret is affected by the value of $\beta$, our proposed algorithms including both fixed and adaptive methods do not require knowing $\beta$.

(b) restricts the suboptimality gap of each action. This is not necessary if the support is bounded. However, with unbounded support, without assumption (b), $\eta^*(\mathbf{x}) - \eta_a(\mathbf{x})$ can increase appropriately with the decrease of $f(\mathbf{x})$, such that the regret of suboptimal action is large, and the identification of best action is hard.

Finally, we clarify notations as follows. Throughout this paper, $\|\cdot\|$ denotes $\ell_2$ norm. $a \lesssim b$ denotes $a \leq Cb$ for some constant $C$, which may depend on the constants in Assumption 2. The notation $\gtrsim$ is defined conversely.

## 4. Minimax Analysis

In this section, we show the minimax lower bound, which characterizes the theoretical limit of regrets of contextual bandits. Throughout this section, denote $\pi : \mathcal{X} \times \mathcal{X}^{t-1} \times \mathbb{R}^{t-1} \to \mathcal{A}$ as the policy, such that each action is selected according to policy $\pi$. To be more precise,

$$A_t = \pi(\mathbf{X}_t; \mathbf{X}_{1:t-1}, Y_{1:t-1}), \tag{4}$$

which indicates that the action $A_t$ at time $t$ depends on the current context and the records of contexts and rewards in previous $t - 1$ steps.

The minimax lower bound for the case with bounded support has been shown in Theorem 4.1 in (Rigollet & Zeevi, 2010). For completeness and notation consistency, we state the results below and provide a simplified proof.

**Theorem 1.** *((Rigollet & Zeevi, 2010), Theorem 4.1) Denote $\mathcal{F}_A$ as the set of pairs $(f, \eta)$ that satisfy Assumption 1 and 2 (which means that the contexts have bounded support). Then*

$$\inf_\pi \sup_{(f,\eta) \in \mathcal{F}_A} R \gtrsim T^{1 - \frac{1+\alpha}{d+2}}. \tag{5}$$

We then show the minimax regret bounds for unbounded support, which is a new result that has not been obtained before.

**Theorem 2.** *Denote $\mathcal{F}_B$ as the set of pairs $(f, \eta)$ that satisfy Assumption 1 and 3 (which means that the contexts have unbounded support). Then*

$$\inf_\pi \sup_{(f,\eta) \in \mathcal{F}_B} R \gtrsim T^{1 - \frac{(\alpha+1)\beta}{\alpha+(d+2)\beta}} + T^{1-\beta}. \tag{6}$$

From the results above, with $\beta \to \infty$, (6) reduces to (5).

*Proof.* (Outline) For bounded support, we just derive the lower bound of regret by analyzing the minimax optimal number of suboptimal actions first. Define

$$S = \mathbb{E}\left[\sum_{t=1}^{T} \mathbf{1}(\eta_{A_t}(\mathbf{X}_t) < \eta^*(\mathbf{X}_t))\right]. \tag{7}$$

$S$ can be lower bounded using standard tools in nonparametric statistics (Tsybakov, 2009), which constructs multiple hypotheses and bounds the minimum error probability. As shown in (Rigollet & Zeevi, 2010), the lower bound of $S$ can then be transformed to the lower bound of $R$.

The minimax analysis becomes more complex with unbounded support. Firstly, the heavy-tailed context distribution requires different hypotheses construction. Secondly, the transformation from the lower bound of $S$ to $R$ does not yield tight lower bounds. We design new approaches to construct a set of candidate functions $\eta$ and derive lower bounds of $R$ directly. □

For bounded context support, regret comes mainly from the region with $\eta^*(\mathbf{x}) - \eta_a(\mathbf{x}) \lesssim T^{-1/(d+2)}$ (which is the classical rate for nonparametric estimation (Tsybakov, 2009)), in which the identification of best action is not guaranteed to be correct. However, with heavy-tailed contexts, regret may also come from the tail, i.e. the region with small $f(\mathbf{x})$, where the number of samples around $\mathbf{x}$ is not enough to yield a reliable best action identification. For heavy-tailed cases, i.e. $\beta$ is small, the regret caused by the tail region may dominate. This also explains why we need to use different techniques to derive the minimax lower bound for heavy-tailed contexts.

In the remainder of this paper, we claim that a method is nearly minimax optimal if the dependence of expected regret on $T$ matches (5) or (6). Following conventions in existing works (Rigollet & Zeevi, 2010; Perchet & Rigollet, 2013; Hu et al., 2020; Gur et al., 2022), currently, the minimax lower bounds are derived for contextual bandit problems with only two actions, thus we do not consider the minimax optimality of regrets with respect to the number of actions $|\mathcal{A}|$.

## 5. Nearest Neighbor Method with Fixed $k$

To begin with, we propose and analyze a simple nearest neighbor method with fixed $k$. We make the following definitions first.

Denote $n_a(t) = |\{i < t | A_i = a\}|$ as the number of steps with action $a$ before time step $t$. Let $\mathcal{N}_t(\mathbf{x}, a)$ be the set of $k$ nearest neighbors among $\{i < t | A_i = a\}$. Define

$$\rho_{a,t}(\mathbf{x}) = \max_{i \in \mathcal{N}_t(\mathbf{x},a)} \|\mathbf{X}_i - \mathbf{x}\| \tag{8}$$

as the $k$ nearest neighbor distance, i.e. the distance from $\mathbf{x}$ to its $k$-th nearest neighbor among all previous steps with action $a$.

With the above notations, we describe the fixed $k$ nearest neighbor method as follows. If $n_a(t) \geq k$, then

$$\hat{\eta}_{a,t}(\mathbf{x}) = \frac{1}{k} \sum_{i \in \mathcal{N}_t(\mathbf{x},a)} Y_i + b + L\rho_{a,t}(\mathbf{x}), \tag{9}$$

in which $b$ has a fixed value

$$b = \sqrt{\frac{2\sigma^2}{k} \ln(dT^{2d+2}|\mathcal{A}|)}, \tag{10}$$

If $n_a(t) < k$, then

$$\hat{\eta}_{a,t}(\mathbf{x}) = \infty. \tag{11}$$

Here we explain our design. If $n_a(t) \geq k$, then it is possible to give a UCB estimate of $\eta_a(\mathbf{x})$, shown in (9). $L\rho_{a,t}(\mathbf{x})$

bounds the estimation bias, while $b$ is an upper bound of the error caused by random noise that holds with high probability. In Lemma 5 in Appendix E, we show that $\hat{\eta}_{a,t}(\mathbf{x})$ is a valid UCB estimate of $\eta_{a,t}(\mathbf{x})$, i.e. $\hat{\eta}_{a,t}(\mathbf{x}) \geq \eta_{a,t}(\mathbf{x})$ holds with high probability. If $n_a(t) < k$, then it is impossible to give a UCB estimate. In this case, we just let $\hat{\eta}_{a,t}(\mathbf{x})$ to be infinite.

Finally, the algorithm selects the action $A_t$ with the maximum UCB value:

$$A_t = \arg\max_a \hat{\eta}_{a,t}(\mathbf{X}_t). \tag{12}$$

According to (11), as long as an action has not been taken for at least $k$ times, the UCB estimate of $\eta_a$ will be infinite. Note that the selection rule (12) ensures that the actions with infinite UCB values will be taken first. Therefore, the first $k|\mathcal{A}|$ steps are used for pure exploration. In this stage, the agent takes each action $a$ for $k$ times. After $k|\mathcal{A}|$ steps, the UCB values for all $\mathbf{x} \in \mathcal{X}$ and $a \in \mathcal{A}$ become finite. Since then, at each step, the action is selected with the maximum UCB value specified in (9).

---

**Algorithm 1** Adaptive nearest neighbor with UCB exploration

---

**for** $t = 1, \ldots, T$ **do**
    Receive context $\mathbf{X}_t$;
    **for** $a \in \mathcal{A}$ **do**
        Calculate $n_a(t) = |\{i < t | A_i = a\}|$;
        **if** $n_a(t) \geq k$ **then**
            Calculate $\hat{\eta}_{a,t}(\mathbf{X}_t)$ using (9);
        **else**
            Let $\hat{\eta}_{a,t}(\mathbf{X}_t) = \infty$;
        **end if**
    **end for**
    $A_t = \arg\max_a \hat{\eta}_{a,t}(\mathbf{X}_t)$;
    Pull $A_t$;
**end for**

---

The procedures above are summarized in Algorithm 1. Compared with (Guan & Jiang, 2018), our method constructs the UCB differently. In (Guan & Jiang, 2018), the UCB is $\frac{1}{k} \sum_{i \in \mathcal{N}_t(\mathbf{x},a)} Y_i + \sigma(T_a(t-1))$, in which $\sigma(T_a(t-1))$ is uniform among all $\mathbf{x}$ with fixed action $a$.

Therefore, the method (Guan & Jiang, 2018) is not adaptive to the suboptimality gap $\eta^*(\mathbf{x}) - \eta_a(\mathbf{x})$. On the contrary, our method has a term $L\rho_{a,t}(\mathbf{x})$ that varies for different $\mathbf{x}$, and thus adapts better to the suboptimality gap. The bound of regret is shown in Theorem 3.

**Theorem 3.** *Under Assumption 1 and 2, the regret of the simple nearest neighbor method with UCB exploration is bounded as follows:*

*(1) If $d > \alpha + 1$, then with $k \sim T^{\frac{2}{d+2}}$,*

$$R \lesssim T^{1-\frac{\alpha+1}{d+2}}|\mathcal{A}|\ln^{\frac{\alpha+1}{2}}(dT^{2d+2}|\mathcal{A}|); \qquad (13)$$

*(2) If $d \leq \alpha + 1$, then with $k \sim T^{\frac{2}{\alpha+3}}$,*

$$R \lesssim T^{\frac{2}{\alpha+3}}|\mathcal{A}|\ln^{\frac{\alpha+1}{2}}(dT^{2d+2}|\mathcal{A}|). \qquad (14)$$

We compare our result with (Guan & Jiang, 2018), which proposes a similar nearest neighbor method. The analysis in (Guan & Jiang, 2018) does not make Tsybakov margin assumption (Assumption 1(a)), and the regret bound is $\tilde{O}(T^{\frac{d+1}{d+2}})$. Without any restriction on $\eta$, Assumption 1(a) holds with $C_\alpha = 1$ and $\alpha = 0$, under which (13) reduces to $\tilde{O}(T^{\frac{d+1}{d+2}})$. Therefore, our result matches (Guan & Jiang, 2018) with $\alpha = 0$. If $\alpha > 0$, which indicates that a small optimality gap only happens with small probability, then the regret of the method in (Guan & Jiang, 2018) is still $\tilde{O}(T^{\frac{d+1}{d+2}})$, while our result improves it to $\tilde{O}(T^{1-\frac{\alpha+1}{d+2}})$. As discussed earlier, compared with (Guan & Jiang, 2018), our method improves the UCB calculation in (17), and is thus more adaptive to the suboptimalilty gap $\eta^*(\mathbf{x}) - \eta_a(\mathbf{x})$. With $\alpha > 0$, our method achieves smaller regret due to a better tradeoff between exploration and exploitation.

Compared with the minimax lower bound shown in Theorem 1, it can be found that the kNN method with fixed $k$ is not completely optimal. With $d > \alpha + 1$, the upper bound matches the lower bound derived in Theorem 1. However, with $d \leq \alpha + 1$, the regret is significantly higher than the minimax lower bound, indicating that there is room for further improvement.

We then analyze the performance for heavy-tailed context distribution. The result is shown in the following theorem.

**Theorem 4.** *Under Assumption 1 and 3, the regret of the simple nearest neighbor method with UCB exploration is bounded as follows:*

$$R \lesssim T^{1-\frac{\beta\min(d,\alpha+1)}{\min(d-1,\alpha)+\max(1,d\beta)+2\beta}}|\mathcal{A}|$$
$$\ln^{\frac{1}{2}\max(d,\alpha+1)}(dT^{2d+2}|\mathcal{A}|). \qquad (15)$$

From (15), there are two phase transitions. The first one is at $d = \alpha + 1$, while the second one is at $d\beta = 1$. Intuitively, the phase transition occurs because the regret is dominated by different regions depending on the settings $\alpha$ and $\beta$. Compared with the minimax lower bound shown in Theorem 2, it can be found that the kNN method with fixed $k$ achieves nearly minimax optimal regret up to logarithm factors if $d > \alpha + 1$ and $\beta > 1/d$, otherwise the regret bound is suboptimal. Here we provide an intuition of the reason why the kNN method with fixed $k$ achieves suboptimal regrets. In the region where the context pdf $f(\mathbf{x})$ is low,

or the suboptimality gap $\eta^*(\mathbf{x}) - \eta_a(\mathbf{x})$ is large, the samples with action $a$ are relatively sparse. In this case, with fixed $k$, the nearest neighbor distances are too large, resulting in a large estimation bias. On the contrary, if $f(\mathbf{x})$ is high or $\eta^*(\mathbf{x}) - \eta_a(\mathbf{x})$ is small, then samples with action $a$ are relatively dense, thus the bias is small, and we can increase $k$ to achieve a better bias and variance tradeoff. Therefore, if $k$ is fixed throughout the support set, then the algorithm estimates the reward function $\eta_a(\mathbf{x})$ in an inefficient way, resulting in suboptimal regrets. Apart from suboptimal regret, another drawback is that with $d \leq \alpha + 1$, the optimal selection of $k$ depends on the margin parameter $\alpha$, which is usually unknown in practice. In the next section, we propose an adaptive nearest neighbor method to address these issues mentioned above.

## 6. Nearest Neighbor Method with Adaptive $k$

In the previous section, we have shown that the standard kNN method with fixed $k$ is suboptimal with $d \leq \alpha + 1$ or $\beta \leq 1/d$. The intuition is that the standard nearest neighbor method does not adjust $k$ based on the pdf and the suboptimality gap. In this section, we propose an adaptive nearest neighbor approach. To achieve a good exploration-exploitation tradeoff and bias-variance tradeoff, $k$ needs to be smaller for small pdf $f(\mathbf{x})$ or large suboptimality gap $\eta^*(x) - \eta_a(x)$, and vice versa. However, as both $f(\mathbf{x})$ and $\eta^*(\mathbf{x}) - \eta_a(\mathbf{x})$ are unknown to the learner, we need to decide $k$ based entirely on existing samples. The guideline of our design is that given a context $\mathbf{X}_t$ at time $t$, we use large $k$ if previous samples are relatively dense around $\mathbf{X}_t$, and vice versa. To be more precise, for all $\mathbf{x} \in \mathcal{X}$, let

$$k_t(\mathbf{x}) = \max\left\{j | L\rho_{a,t,j}(\mathbf{x}) \leq \sqrt{\frac{\ln T}{j}}\right\}, \qquad (16)$$

in which $\rho_{a,t,j}(\mathbf{x})$ is the distance from $x$ to its $j$-th nearest neighbors among existing samples with action $a$, i.e. $\{i < t | A_i = a\}$. Such selection of $k$ makes the bias term $L\rho_{a,t,j}(\mathbf{x})$ matches the variance term $\sqrt{\ln T/j}$, thus (16) achieves a good tradeoff between bias and variance. The exploration-exploitation tradeoff is also desirable as $\rho_{a,t,j}(\mathbf{x})$ is large with large $\eta^*(\mathbf{x}) - \eta_a(\mathbf{x})$, which yields smaller $k$. Note that (16) can be calculated only if $L\rho_{a,t,1} \leq \sqrt{\ln T}$, which means that the 1-nearest neighbor distance can not be too large. At some time step $t$, for some action $a$, if there is no existing samples, or $\mathbf{X}_t$ is more than $\sqrt{\ln T}/L$ far away from any existing samples $\mathbf{X}_1, \ldots, \mathbf{X}_{t-1}$, then we can just let the UCB estimate to be infinite, i.e. $\hat{\eta}_{a,t}(\mathbf{x}) = \infty$. Otherwise, we calculate the upper confidence bound as follows:

$$\hat{\eta}_{a,t}(\mathbf{x}) = \frac{1}{k_{a,t}(\mathbf{x})}\sum_{i\in\mathcal{N}_t(\mathbf{x},a)}Y_i + b_{a,t}(\mathbf{x}) + L\rho_{a,t}(\mathbf{x}), \quad (17)$$

in which $\mathcal{N}_t(\mathbf{x}, a)$ is the set of $k_{a,t}(\mathbf{x})$ neighbors of $x$ among $\{i < t | A_i = a\}$, $\rho_{a,t}(\mathbf{x})$ is the corresponding $k_{a,t}(\mathbf{x})$ neighbor distance of $\mathbf{x}$, i.e. $\rho_{a,t}(\mathbf{x}) = \rho_{a,t,k_{a,t}(\mathbf{x})}(\mathbf{x})$, and

$$b_{a,t}(\mathbf{x}) = \sqrt{\frac{2\sigma^2}{k_{a,t}(\mathbf{x})} \ln(dT^{2d+3}|\mathcal{A}|)}. \qquad (18)$$

Similar to the fixed nearest neighbor method, the last two terms in (17) cover the uncertainty of reward function estimation. The term $b_{a,t}(\mathbf{x})$ gives a high probability bound of random error, and $L\rho_{a,t}(\mathbf{x})$ bounds the bias. With the UCB calculation in (17), the $\hat{\eta}_{a,t}(\mathbf{x})$ is an upper bound of $\eta(\mathbf{x})$ that holds with high probability, so that the exploration and exploitation can be balanced well. The complete description of the newly proposed adaptive nearest neighbor method is shown in Algorithm 2.

---

**Algorithm 2** Adaptive nearest neighbor with UCB exploration

---

**for** $t = 1, \ldots, T$ **do**
    Receive context $\mathbf{X}_t$;
    **for** $a \in \mathcal{A}$ **do**
        **if** $L\rho_{a,t,1}(\mathbf{X}_t) > \sqrt{\ln T}$ **then**
            $\hat{\eta}_{a,t}(\mathbf{X}_t) = \infty$;
        **else**
            Calculate $k_t(\mathbf{X}_t)$ using (16);
            Calculate $\hat{\eta}_{a,t}(\mathbf{X}_t)$ using (17);
        **end if**
    **end for**
    $A_t = \arg\max_a \hat{\eta}_{a,t}(\mathbf{X}_t)$;
    Pull $A_t$;
**end for**

---

We then analyze the regret of the adaptive method for both bounded and unbounded supports of contexts.

**Theorem 5.** *Under Assumption 1 and 2, the regret of the adaptive nearest neighbor method with UCB exploration is bounded by*

$$R \lesssim T|\mathcal{A}| \left( \frac{T}{\ln T} \right)^{-\frac{1+\alpha}{d+2}}. \qquad (19)$$

By comparing Theorem 5 with Theorem 3, it can be found that for the case with bounded support, the adaptive method improves over the fixed $k$ nearest neighbor method. From the minimax bound in Theorem 1, the fixed $k$ method is only optimal for $d \geq \alpha + 1$, while the adaptive method is also optimal for $d < \alpha + 1$, up to logarithm factors. An intuitive explanation is that with large $\alpha$, the suboptimality gap $\eta^*(\mathbf{x}) - \eta_a(\mathbf{x})$ is small only in a small region, and the exploration-exploitation tradeoff becomes harder, thus the advantage of the adaptive method over the fixed one becomes more obvious.

We then analyze the performance of the adaptive nearest neighbor method for heavy-tailed distribution. The result is shown in the following theorem.

**Theorem 6.** *Under Assumption 1 and Assumption 3, the regret of the adaptive nearest neighbor method with UCB exploration is bounded by*

$$R \lesssim \begin{cases} T^{1-\min\left\{\frac{(\alpha+1)\beta}{\alpha+(d+2)\beta}, \beta\right\}} |\mathcal{A}| \ln T & if \quad \beta \neq \frac{1}{d+2} \\ T^{\frac{d+2}{d+1}} |\mathcal{A}| \ln^2 T & if \quad \beta = \frac{1}{d+2}. \end{cases} \qquad (20)$$

Compared with the minimax lower bound shown in Theorem 2, it can be found that our method achieves nearly minimax optimal regret up to a logarithm factor. Regarding this result, we have some additional remarks.

**Remark 1.** *It can be found that with $\beta \to \infty$, the regret bound in (20) reduces to (19). As discussed earlier, the case that contexts have bounded support and $f$ is bounded away from zero can be viewed as a special case with $\beta \to \infty$.*

**Remark 2.** *In (Zhao & Lai, 2021b), it is shown that the optimal rate of the excess risk of nonparametric classification is $\tilde{O}\left(N^{-\frac{(\alpha+1)\beta}{\alpha+(d+2)\beta}}\right)^2$. From (20), the average regret over all $T$ steps is $\tilde{O}\left(T^{-\frac{(\alpha+1)\beta}{\alpha+(d+2)\beta}}\right)$, which has the same rate as the nonparametric classification problem.*

## 7. Numerical Experiments

To begin with, to validate our theoretical analysis, we run experiments using some synthesized data. We then move on to experiments with the MNIST dataset (LeCun, 1998).

### 7.1. Synthesized Data

To begin with, we conduct experiments with $d = 1$. In each experiment, we run $T = 1,000$ steps and compare the performance of the adaptive nearest neighbor method with the UCBogram (Rigollet & Zeevi, 2010), ABSE (Perchet & Rigollet, 2013) and fixed $k$ nearest neighbor method. For a fair comparison, for UCBogram and ABSE, we try different numbers of bins and only pick the one with the best performance. The results are shown in Figure 1. In (a), (b), (c), and (d), the contexts follow uniform distribution in $[-1, 1]$, standard Gaussian distribution, $t_4$ distribution, and Cauchy distribution, respectively. The uniform distribution is an example of distributions with bounded support. The Gaussian, $t_4$ and Cauchy distribution satisfy the tail assumption (Assumption 3(a)) with $\beta = 1$, $0.8$ and $0.5$, respectively. In each experiment, there are two actions. For uniform and Gaussian distribution, we have $\eta_1(x) = x$ and $\eta_2(x) = -x$.

---

[2]The analysis in (Zhao & Lai, 2021b) is under a general smoothness assumption with parameter $p$. $p = 1$ corresponds to the Lipschitz assumption (Assumption 1(c) in this paper). Therefore, here we replace the bounds in (Zhao & Lai, 2021b) with $p = 1$.

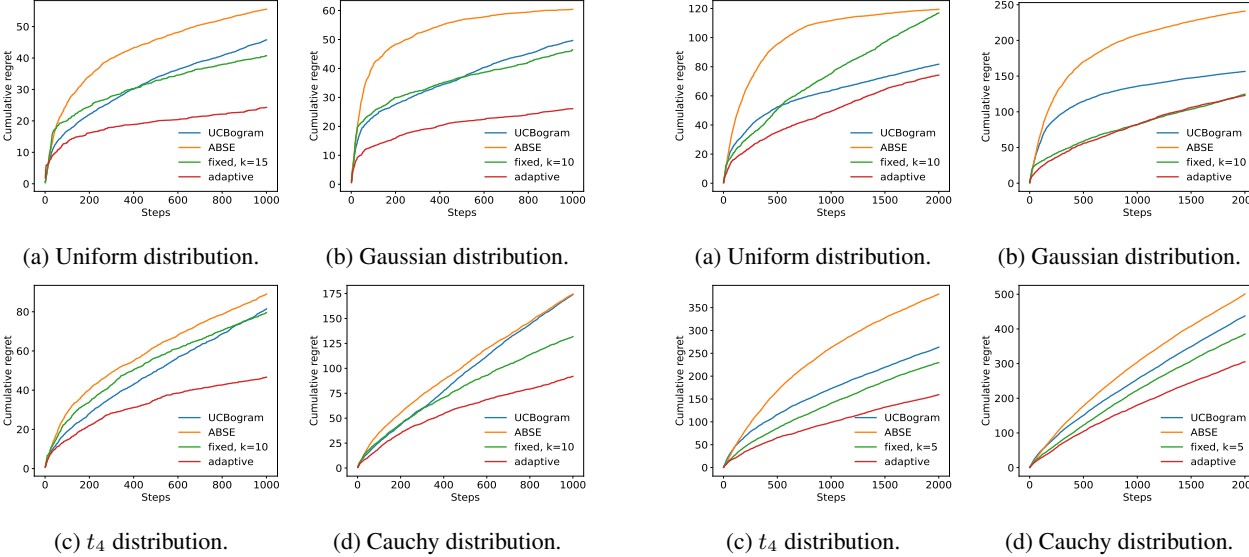

*Figure 1.* Comparison of cumulative regrets of different methods for one dimensional distributions.

*Figure 2.* Comparison of cumulative regrets of different methods for one dimensional distributions.

For $t_4$ and Cauchy distribution, since they are heavy-tailed, to ensure that Assumption 3(b) is satisfied, we do not use the linear reward function. Instead, we let $\eta_1(x) = \sin(x)$ and $\eta_2(x) = \cos(x)$. To make the comparison more reliable, the values in each curve in Figure 1 are averaged over $m = 100$ random and independent trials.

We then run experiments for two dimensional distributions. In these experiments, the context distributions are just Cartesian products of two one dimensional distributions. The two dimensional Gaussian distribution still satisfies Assumption 3(a) with $\beta = 1$, and the two dimensional $t_4$ and Cauchy distribution satisfy Assumption 3(a) with $\beta = 2/3$ and $1/3$, respectively, which are lower than the one dimensional case. The results are shown in Figure 2.

From these experiments, it can be observed that the adaptive nearest neighbor method significantly outperforms the other baselines.

### 7.2. Real Data

Now we run experiments using the MNIST dataset (LeCun, 1998), which contains $60,000$ images of handwritten digits with size $28 \times 28$. Following the settings in (Guan & Jiang, 2018), the images are regarded as contexts, and there are 10 actions from 0 to 9. The reward is 1 if the selected action equals the true label, and 0 otherwise. The results are shown in Figure 3. Image data have high dimensionality but low intrinsic dimensionality. Compared with bin splitting based methods (Rigollet & Zeevi, 2010; Perchet & Rigollet, 2013), nearest neighbor methods are more adaptive to local intrinsic dimension (Kpotufe, 2011). Therefore, in this experiment, we do not compare with the bin splitting based methods.

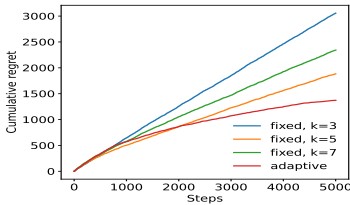

*Figure 3.* Cumulative regrets for MNIST dataset.

From Figure 3, the adaptive kNN method performs better than the standard kNN method with various values of $k$.

## 8. Conclusion

This paper analyzes the contextual bandit problem that allows the context distribution to be heavy-tailed. To begin with, we have derived the minimax lower bound of the expected cumulative regret. We then show that the expected cumulative regret of the fixed $k$ nearest neighbor method is suboptimal compared with the minimax lower bound. To close the gap, we have proposed an adaptive nearest neighbor approach, which significantly improves the performance, and the bound of expected regret matches the minimax lower bound up to logarithm factors. Finally, we have conducted numerical experiments to validate our results.

In the future, this work can be extended in the following ways. Firstly, following existing analysis in (Gur et al., 2022), it may be meaningful to design a smoothness adaptive method that can handle any Hölder smoothness parameters. Secondly, it is worth extending current work to handle dynamic regret functions. Finally, the theories developed in this paper can be extended to more complicated tasks, such as reinforcement learning (Zhao & Lai, 2024).

## Impact Statement

This paper presents work whose goal is to advance the field of Machine Learning. There are many potential societal consequences of our work, none which we feel must be specifically highlighted here.

## Acknowledgements

This work is supported by STI 2030—Major Projects (No. 2021ZD0201405), Shenzhen Basic Research Project (Natural Science Foundation) Basic Research Key Project (NO. JCYJ20241202124430041), 2025 Open Project of the Key Laboratory of Blockchain Technology and Data Security of the Ministry of Industry and Information Technology, Open Research Fund from Guangdong Laboratory of Artificial Intelligence and Digital Economy (SZ) (NO. GML-KF-24-23), National Natural Science Foundation of China (No.62372120, 62102108), GuangDong Basic and Applied Basic Research Foundation (No.2022A1515010061), and Science and Technology Projects in Guangzhou (No.2025A03J3182).

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

## A. Examples of Heavy-tailed Distributions

This section explains Example 1 and 2 in the paper. For Example 1,

$$P(f(\mathbf{X}) < t) = \int_{\mathcal{X}} f(\mathbf{x})\mathbf{1}(f(\mathbf{x}) < t)d\mathbf{x} \le tV(\mathcal{X}). \tag{21}$$

For Example 2, from Hölder's inequality,

$$\int f^{1-\beta}(\mathbf{x})d\mathbf{x} = \int f^{1-\beta}(\mathbf{x})(1 + \|\mathbf{x}\|^{\gamma})\frac{1}{1 + \|\mathbf{x}\|^{\gamma}}$$
$$\le \left(\int f(\mathbf{x})(1 + \|\mathbf{x}\|^{\gamma})^{\frac{1}{1-\beta}}d\mathbf{x}\right)^{1-\beta}\left(\int (1 + \|\mathbf{x}\|^{\gamma})^{-\frac{1}{\beta}}d\mathbf{x}\right)^{\beta}. \tag{22}$$

Let $\gamma = p(1 - \beta)$, then $\left(\int f(\mathbf{x})(1 + \|\mathbf{x}\|^{\gamma})^{\frac{1}{1-\beta}}d\mathbf{x}\right)^{1-\beta} < \infty$. If $\beta < p/(p + d)$, then $\gamma/\beta > d$, thus $\left(\int(1 + \|\mathbf{x}\|^{\gamma})^{-\frac{1}{\beta}}d\mathbf{x}\right)^{\beta} < \infty$. Hence $\int f^{1-\beta}(\mathbf{x})d\mathbf{x} < \infty$, and

$$P(f(\mathbf{X}) < t) = P(f^{-\beta}(\mathbf{X}) > t^{-\beta}) \le t^{\beta}\mathbb{E}[f^{-\beta}(\mathbf{X})] = t^{\beta}\int f^{1-\beta}(\mathbf{x})d\mathbf{x}. \tag{23}$$

Therefore for all $\beta < p/(p + d)$, Assumption 3(a) holds with some finite $C_{\beta}$.

For subgaussian or subexponential random variables, $\mathbb{E}[\|\mathbf{X}\|^p] < \infty$ holds for any $p$, thus Assumption 3(a) holds for $\beta$ arbitrarily close to 1.

## B. Expected Sample Density

In this section, we define *expected sample density*, which is then used in the later analysis. Throughout this section, denote $x(j)$ as the value of $j$-th component of vector $\mathbf{x}$.

**Definition 1.** *(expected sample density) $q_a : \mathcal{X} \to \mathbb{R}$ is defined as the function such that for all $S \subseteq \mathcal{X}$,*

$$\mathbb{E}\left[\sum_{t=1}^{T}\mathbf{1}(\mathbf{X}_t \in S, A_t = a)\right] = \int_{S} q_a(\mathbf{x})d\mathbf{x}. \tag{24}$$

To show the existence of $q_a$, define

$$Q_a(\mathbf{x}) = \mathbb{E}\left[\sum_{t=1}^{T}\mathbf{1}(\mathbf{X}_t \in \{u|u(1) \le x(1), \ldots, u(d) \le x(d)\}, A_t = a)\right]. \tag{25}$$

Then let

$$q_a(\mathbf{x}) = \frac{\partial^d Q_a}{\partial x(1)\ldots\partial x(d)}\bigg|_x, \tag{26}$$

and then (24) is satisfied for all $S \subseteq \mathcal{X}$.

Then we show the following basic lemmas.

**Lemma 1.** *Regardless of $\eta$, $q_a$ satisfies*

$$q_a(\mathbf{x}) \le Tf(\mathbf{x}) \tag{27}$$

*for almost all $\mathbf{x} \in \mathcal{X}$.*

*Proof.* Note that $P(\mathbf{X}_t \in S) = \int_S f(\mathbf{x})d\mathbf{x}$. Therefore for all set $S$,

$$\mathbb{E}\left[\sum_{t=1}^{T} \mathbf{1}(\mathbf{X}_t \in S, A_t = a)\right] \leq T \int_S f(\mathbf{x})d\mathbf{x}. \tag{28}$$

From (24) and (28), $\int_S q_a(\mathbf{x})d\mathbf{x} \leq T \int_S f(\mathbf{x})d\mathbf{x}$ for all $S$. Therefore $q_a(\mathbf{x}) \leq Tf(\mathbf{x})$ for almost all $\mathbf{x} \in \mathcal{X}$.  □

**Lemma 2.** $R = \sum_{a \in \mathcal{A}} R_a$, in which $R_a$ is defined as

$$R_a := \int_{\mathcal{X}} (\eta^*(\mathbf{x}) - \eta_a(\mathbf{x}))q_a(\mathbf{x})d\mathbf{x}. \tag{29}$$

*Proof.*

$$
\begin{aligned}
R &= \mathbb{E}\left[\sum_{t=1}^{T} (\eta^*(\mathbf{X}_t) - \eta_{A_t}(\mathbf{X}_t))\right] \\
&= \sum_{a \in \mathcal{A}} \mathbb{E}\left[\sum_{t=1}^{T} (\eta^*(\mathbf{X}_t) - \eta_a(\mathbf{X}_t))\mathbf{1}(A_t = a)\right] \\
&= \sum_{a \in \mathcal{A}} \int_{\mathcal{X}} (\eta^*(\mathbf{x}) - \eta_a(\mathbf{x}))\, q_a(\mathbf{x})d\mathbf{x}.
\end{aligned}
\tag{30}
$$

The proof is complete.  □

## C. Proof of Theorem 1

Recall that

$$R = \mathbb{E}\left[\sum_{t=1}^{T} (\eta^*(\mathbf{X}_t) - \eta_{A_t}(\mathbf{X}_t))\right]. \tag{31}$$

Now we define

$$S = \mathbb{E}\left[\sum_{t=1}^{T} \mathbf{1}(\eta_{A_t}(\mathbf{X}_t) < \eta^*(\mathbf{X}_t))\right] \tag{32}$$

as the expected number of steps with suboptimal actions.

The following lemma characterizes the relationship between $S$ and $R$.

**Lemma 3.** *There exists a constant $C_0$, such that*

$$R \geq C_0 S^{\frac{\alpha+1}{\alpha}} T^{-\frac{1}{\alpha}}. \tag{33}$$

*Proof.* The proof of Lemma 3 follows the proof of Lemma 3.1 in (Rigollet & Zeevi, 2010). For completeness and consistency of notations, we show the proof in Appendix I.10.  □

From now on, we only discuss the case with only two actions, such that $\mathcal{A} = \{1, -1\}$. Construct $B$ disjoint balls with centers $\mathbf{c}_1, \ldots, \mathbf{c}_B$ and radius $h$. Let

$$f(\mathbf{x}) = \sum_{j=1}^{B} \mathbf{1}(\mathbf{x} \in B_j), \tag{34}$$

in which $B_j = \{\mathbf{x}' \,|\, \|\mathbf{x}' - \mathbf{c}_j\| \leq h\}$ is the $j$-th ball. To ensure that the pdf defined above is normalized (i.e. $\int f(\mathbf{x})d\mathbf{x} = 1$), $B$ and $h$ need to satisfy

$$Bv_d h^d = 1, \tag{35}$$

in which $v_d$ is the volume of $d$ dimensional unit ball.

Let $\eta_1(\mathbf{x}) = \eta(\mathbf{x})$ and $\eta_2(\mathbf{x}) = 0$, with

$$\eta(\mathbf{x}) = \sum_{j=1}^{K} v_j h \mathbf{1}(\mathbf{x} \in B(c_j, h)), \tag{36}$$

in which $v_j \in \{-1, 1\}$ for $j = 1, \ldots, K$. To satisfy the margin assumption (Assumption 1(a)), note that

$$\mathrm{P}(0 < |\eta(\mathbf{X})| \le t) \le \begin{cases} K v_d h^d & \text{if} \quad t \ge h \\ 0 & \text{if} \quad t < h. \end{cases} \tag{37}$$

Note that for any suboptimal action $a$, $\eta^*(\mathbf{x}) - \eta_a(\mathbf{x}) = |\eta(\mathbf{x})|$. Assumption 1(a) requires that $\mathrm{P}(0 < |\eta(\mathbf{X})| \le t) \le C_\alpha t^\alpha$. Therefore, it suffices to ensure that

$$K v_d h^d = C_\alpha h^\alpha. \tag{38}$$

Then

$$\begin{aligned} S &= \sum_{j=1}^{K} \sum_{t=1}^{T} \mathrm{P}(\mathbf{X}_t \in B_j, A_t \ne a^*(\mathbf{X}_t)) \\ &\ge \sum_{j=1}^{K} \sum_{t=1}^{T} \int_{B_j} f(\mathbf{x}) \mathrm{P}(A_t \ne a^*(\mathbf{x}) | \mathbf{X}_t = x) d\mathbf{x} \\ &= \sum_{j=1}^{K} \sum_{t=1}^{T} \int_{B_j} f(\mathbf{x}) \mathrm{P}(A_t \ne v_j | \mathbf{X}_t = x) d\mathbf{x} \\ &= \sum_{j=1}^{k} \sum_{t=1}^{T} \mathbb{E} \left[ \int_{B_j} f(\mathbf{x}) \mathbf{1}(\pi(x; \mathbf{X}_{1:t-1}, Y_{1:t-1}) \ne v_j) d\mathbf{x} \right]. \end{aligned} \tag{39}$$

Define

$$\hat{v}_j(t) = \mathrm{sign} \left( \int_{B_j} f(\mathbf{x}) \pi(x; \mathbf{X}_{1:t-1}, Y_{1:t-1}) d\mathbf{x} \right). \tag{40}$$

Then

$$\int_{B_j} f(\mathbf{x}) \mathbf{1}(\pi(x; \mathbf{X}_{1:t-1}, Y_{1:t-1}) - \hat{v}_j(t)) d\mathbf{x} \ge \int_{B_j} f(\mathbf{x}) \mathbf{1}\left(\pi(x; \mathbf{X}_{1:t-1}, Y_{1:t-1}) = -\hat{v}_j(t)\right) d\mathbf{x}. \tag{41}$$

Since

$$\begin{aligned} &\int_{B_j} f(\mathbf{x}) \mathbf{1}\left(\pi(x; \mathbf{X}_{1:t-1}, Y_{1:t-1}) - \hat{v}_j(t)\right) d\mathbf{x} + \int_{B_j} f(\mathbf{x}) \mathbf{1}\left(\pi(x; \mathbf{X}_{1:t-1}, Y_{1:t-1}) = -\hat{v}_j(t)\right) d\mathbf{x} \\ &= \int_{B_j} f(\mathbf{x}) d\mathbf{x}, \end{aligned} \tag{42}$$

we have

$$\int_{B_j} f(\mathbf{x}) \mathbf{1}\left(\pi(x; \mathbf{X}_{1:t-1}, Y_{1:t-1}) = \hat{v}_j(t)\right) d\mathbf{x} \ge \frac{1}{2} \int_{B_j} f(\mathbf{x}) d\mathbf{x}. \tag{43}$$

If $\hat{v}_j(t) \ne v_j$, then

$$\int_{B_j} f(\mathbf{x}) \mathbf{1}\left(\pi(x; \mathbf{X}_{1:t-1}, Y_{1:t-1}) \ne v_j\right) d\mathbf{x} \ge \frac{1}{2} \int_{B_j} f(\mathbf{x}) d\mathbf{x}. \tag{44}$$

Therefore, from (39),

$$
\begin{aligned}
S & \geq \sum_{j=1}^{K}\sum_{t=1}^{T}\frac{1}{2}\mathrm{P}(\hat{v}_j(t) \neq v_j)\int_{B_j} f(\mathbf{x})d\mathbf{x} \\
& \geq \sum_{j=1}^{K}\sum_{t=1}^{T}\frac{1}{2}v_d h^d \mathrm{P}(\hat{v}_j(t) \neq v_j).
\end{aligned}
\tag{45}
$$

Note that the error probability of hypothesis testing between distance $p$ and $q$ is at least $(1 - \mathbb{TV}(p, q))/2$, in which $\mathbb{TV}$ denotes the total variation distance. Let $(V_1, \ldots, V_K)$ be a vector of $K$ random variables taking values from $\{-1, 1\}^K$ randomly. In other words, $\mathrm{P}(V_j = 1) = \mathrm{P}(V_j = -1) = 1/2$, and $V_j$ for different $j$ are i.i.d. Denote $\mathbb{P}_{XY|V_j=v_j}$ as the distribution of $X$ and $Y$ conditional on $V_j = v_j$. Moreover, $\mathbb{P}_{XY|V_j=v_j}^{t-1}$ means the distribution of $X$ and $Y$ of the first $t-1$ samples conditional on $V_j = v_j$. Then

$$
\begin{aligned}
\mathrm{P}(\hat{v}_j(t) \neq v_j) & \geq \frac{1}{2}\left(1 - \mathbb{TV}\left(\mathbb{P}_{XY|v_j=1}^{t-1}||\mathbb{P}_{XY|V_j=-1}^{t-1}\right)\right) \\
& \geq \frac{1}{2}\left(1 - \sqrt{\frac{1}{2}D\left(\mathbb{P}_{XY|V_j=1}^{t-1}||\mathbb{P}_{XY|V_j=-1}^{t-1}\right)}\right),
\end{aligned}
\tag{46}
$$

in which the second step uses Pinsker's inequality (Fedotov et al., 2003), and $D(p||q)$ denotes the Kullback-Leibler (KL) divergence between distributions $p$ and $q$. Note that the KL divergence between the conditional distribution is bounded by

$$
D(\mathbb{P}_{Y|X,V_j=1}||\mathbb{P}_{Y|X,V_j=-1}) \leq \frac{1}{2}(\eta_1(\mathbf{x}) - \eta_2(\mathbf{x}))^2 \leq \frac{1}{2}\eta^2(\mathbf{x}) = \frac{1}{2}h^2
\tag{47}
$$

for $\mathbf{x} \in B_j$. Therefore

$$
\begin{aligned}
D(\mathbb{P}_{XY|V_j=1}||\mathbb{P}_{XY|V_j=-1}) & = \int f(\mathbf{x})D(\mathbb{P}_{Y|X,V_j=1}||\mathbb{P}_{Y|X,V_j=-1})d\mathbf{x} \\
& \leq \int_{B_j}\frac{1}{2}h^2 d\mathbf{x} \\
& = \frac{1}{2}v_d h^{d+2}.
\end{aligned}
\tag{48}
$$

Hence, from (46),

$$
\begin{aligned}
\mathrm{P}(\hat{v}_j(t) \neq v_j) & \geq \frac{1}{2}\left(1 - \sqrt{\frac{1}{4}(t-1)v_d h^{d+2}}\right) \\
& \geq \frac{1}{2}\left(1 - \frac{1}{2}\sqrt{T v_d h^{d+2}}\right).
\end{aligned}
\tag{49}
$$

Recall (45),

$$
\begin{aligned}
S & \geq \frac{1}{4}\sum_{j=1}^{K}\sum_{t=1}^{T}v_d h^d\left(1 - \frac{1}{2}\sqrt{T v_d h^{d+2}}\right) \\
& = \frac{1}{4}KT v_d h^d\left(1 - \frac{1}{2}\sqrt{T v_d h^{d+2}}\right) \\
& = \frac{1}{4}C_\alpha T h^\alpha\left(1 - \frac{1}{2}\sqrt{T v_d h^{d+2}}\right),
\end{aligned}
\tag{50}
$$

in which the last step comes from (38).

Let $h \sim T^{-\frac{1}{d+2}}$, then

$$
S \gtrsim T^{1 - \frac{\alpha}{d+2}}.
\tag{51}
$$

From Lemma 3,

$$
\begin{aligned}
R & \gtrsim T^{\left(1-\frac{\alpha}{d+2}\right)\frac{\alpha+1}{\alpha}} T^{-\frac{1}{\alpha}} \sim T^{1-\frac{\alpha+1}{d+2}} \\
& \sim T^{1-\frac{\alpha+1}{d+2}}.
\end{aligned}
\tag{52}
$$

## D. Proof of Theorem 2

In this section, we derive the minimax lower bound of the expected regret with unbounded support. Recall that in the case with bounded support of contexts (Proof of Theorem 1 in Appendix C), we construct $B$ disjoint balls with pdf $f(\mathbf{x}) = 1$ for all $\mathbf{x} \in \mathcal{X}$. Now for the case with unbounded support, the distribution of context has tails, on which the pdf $f(\mathbf{x})$ is small. Therefore, we modify the construction of balls as follows. We now construct $B + 1$ disjoint balls with center $\mathbf{c}_0, \dots, \mathbf{c}_B$, such that

$$
\begin{aligned}
B_0 & = \{\mathbf{x}' | \|\mathbf{x}' - \mathbf{c}_0\| \le r_0\}, \tag{53} \\
B_j & = \{\mathbf{x}' | \|\mathbf{x}' - \mathbf{c}_j\| \le h\}. \tag{54}
\end{aligned}
$$

Let

$$
\eta(\mathbf{x}) = \sum_{j=1}^{K} v_j h \mathbf{1}(\mathbf{x} \in B(\mathbf{c}_j, h)),
\tag{55}
$$

in which $v_j \in \{0, 1\}$ is unknown, and

$$
f(\mathbf{x}) = \mathbf{1}(\mathbf{x} \in B_0) + \sum_{j=1}^{B} m \mathbf{1}(\mathbf{x} \in B_j),
\tag{56}
$$

in which $m \ll 1$ will be determined later. Here we construct one ball that denotes the center region which has the most of probability mass, as well as $B$ balls that denotes the tail region. For simplicity, we let $\eta(\mathbf{x}) = 0$ at the largest ball $B_0$, and only

To satisfy the margin condition (i.e. Assumption 1(a)), note that now

$$
P(0 < |\eta(\mathbf{X})| < t) \le \begin{cases} mKv_d h^d & \text{if} \quad t > h \\ 0 & \text{if} \quad t \le h. \end{cases}
\tag{57}
$$

The right hand side of (57) can not exceed $C_\alpha t^\alpha$, which requires

$$
mKv_d h^d \le C_\alpha h^\alpha.
\tag{58}
$$

Moreover, to satisfy the tail assumption (Assumption 3(a)), note that

$$
P(f(\mathbf{X}) < t) \le \begin{cases} mKv_d h^d & \text{if} \quad t > m \\ 0 & \text{if} \quad t \le m. \end{cases}
\tag{59}
$$

The right hand side of (59) can not exceed $C_\beta t^\beta$, which requires

$$
mKv_d h^d \lesssim C_\beta m^\beta.
\tag{60}
$$

Following (45), $S$ can be lower bounded by

$$
\begin{aligned}
S & \ge \sum_{j=1}^{K} \sum_{t=1}^{T} \frac{1}{2} P(\hat{v}_j(t) \ne v_j) \int_{B_j} f(\mathbf{x}) d\mathbf{x} \\
& = \sum_{j=1}^{K} \sum_{t=1}^{T} \frac{1}{2} m v_d h^d P(\hat{v}_j(t) \ne v_j) \\
& \ge \frac{1}{4} \sum_{j=1}^{K} \sum_{t=1}^{T} m v_d h^d \left(1 - \sqrt{\frac{1}{2} D(\mathbb{P}_{XY|V_j=1} \| \mathbb{P}_{XY|V_j=-1})}\right) \\
& \ge \frac{1}{4} \sum_{j=1}^{K} \sum_{t=1}^{T} m v_d h^d \left(1 - \frac{1}{2} \sqrt{T m v_d h^{d+2}}\right).
\end{aligned}
\tag{61}
$$

From (61), we pick $m$ and $h$ to ensure that

$$Tmv_dh^{d+2} < \frac{1}{2}. \tag{62}$$

Then under three conditions (58), (60) and (62),

$$S \gtrsim KTmh^d. \tag{63}$$

It remains to determine the value of $m$, $h$ and $K$ based on these three conditions. Let

$$h \sim (Tm)^{-\frac{1}{d+2}}, \tag{64}$$

$$m \sim T^{-\frac{\alpha}{\alpha+\beta(d+2)}}, \tag{65}$$

and

$$K \sim h^{\alpha-d}/m, \tag{66}$$

then

$$S \gtrsim Th^\alpha \sim T^{1-\frac{\alpha\beta}{\alpha+\beta(d+2)}}. \tag{67}$$

Based on Lemma 3,

$$R \gtrsim S^{\frac{1+\alpha}{\alpha}}T^{-\frac{1}{\alpha}} \sim T^{1-\frac{(\alpha+1)\beta}{\alpha+\beta(d+2)}}. \tag{68}$$

It remains to show that $R \gtrsim T^{1-\beta}$. Let $h \sim 1$, $K \sim T^{1-\beta}$ and $m \sim 1/T$, the conditions (58), (60) and (62) are still satisfied. In this case,

$$S \gtrsim T^{1-\beta}. \tag{69}$$

Direct transformation using Lemma 3 yields suboptimal bound. Intuitively, for the case with heavy tails (i.e. $\beta$ is small), the regret mainly occur at the tail of the context distribution. Therefore, we bound the expected regret again.

$$\begin{aligned}
R &= \mathbb{E}\left[\sum_{t=1}^{T}(\eta^*(\mathbf{X}_t) - \eta_{A_t}(\mathbf{X}_t))\right] \\
&\stackrel{(a)}{=} \mathbb{E}\left[\sum_{t=1}^{T}h\mathbf{1}\left(\eta_{A_t}(\mathbf{X}_t) < \eta^*(\mathbf{X}_t)\right)\right] \\
&\stackrel{(b)}{=} S \gtrsim T^{1-\beta}.
\end{aligned} \tag{70}$$

(a) comes from the construction of $\eta$ in (55). (b) holds since we set $h = 1$ here.

Combine (68) and (70),

$$R \gtrsim T^{1-\frac{(\alpha+1)\beta}{\alpha+\beta(d+2)}} + T^{1-\beta}. \tag{71}$$

## E. Proof of Theorem 3

To begin with, we show the following lemma.

**Lemma 4.** *For all $u > 0$,*

$$P\left(\sup_{x,a}\left|\frac{1}{k}\sum_{i\in\mathcal{N}_t(\mathbf{x},a)}W_i\right| > u\right) \le dT^{2d}|\mathcal{A}|e^{-\frac{ku^2}{2\sigma^2}}. \tag{72}$$

*Proof.* The proof is shown in Appendix I.1. □

From Lemma 4, recall the definition of $b$ in (10),

$$P\left(\sup_{x,a}\left|\frac{1}{k}\sum_{i\in\mathcal{N}_t(\mathbf{x},a)}W_i\right|>b\right)\leq\frac{1}{T^2}. \tag{73}$$

Therefore, with probability $1-1/T$, for all $\mathbf{x}\in\mathcal{X}$, $a\in\mathcal{A}$ and $t=1,\ldots,T$, $|\sum_{i\in\mathcal{N}_t(\mathbf{x},a)}W_i|/k\leq b$. Denote $E$ as the event such that $|\sum_{i\in\mathcal{N}_t(\mathbf{x},a)}W_i|/k\leq b$, $\forall x,a,t$, then

$$
\begin{aligned}
P(E) &= P\left(\cap_{t=1}^T\left\{\sup_{x,a}\left|\frac{1}{k}\sum_{i\in\mathcal{N}_t(\mathbf{x},a)}W_i\right|\leq b\right\}\right)\\
&= 1-P\left(\cap_{t=1}^T\left\{\sup_{x,a}\left|\frac{1}{k}\sum_{i\in\mathcal{N}_t(\mathbf{x},a)}W_i\right|>b\right\}\right)\\
&\geq 1-T\frac{1}{T^2}\geq 1-\frac{1}{T}.
\end{aligned}
\tag{74}
$$

Recall the calculation of UCB in (9). Based on Lemma 4, we then show some properties of the UCB in (9).

**Lemma 5.** *Under E, if $|\{i<t|A_i=a\}|\geq k$, then*

$$\eta_a(t)\leq\hat{\eta}_{a,t}(\mathbf{x})\leq\eta_a(\mathbf{x})+2b+2L\rho_{a,t}(\mathbf{x}). \tag{75}$$

*Proof.* The proof is shown in Appendix I.2. □

We then bound the number of steps with suboptimal action $a$. Define

$$n(x,a,r):=\sum_{t=1}^T\mathbf{1}\left(\|\mathbf{X}_t-\mathbf{x}\|<r,A_t=a\right). \tag{76}$$

Then the following lemma holds.

**Lemma 6.** *Under E, for any $\mathbf{x}\in\mathcal{X}$, $a\in\mathcal{A}$, if $\eta^*(\mathbf{x})-\eta_a(\mathbf{x})>2b$, define*

$$r_a(\mathbf{x})=\frac{\eta^*(\mathbf{x})-\eta_a(\mathbf{x})-2b}{6L}, \tag{77}$$

*then*

$$n(x,a,r_a(\mathbf{x}))\leq k. \tag{78}$$

*Proof.* The proof is shown in Appendix I.3. □

From Lemma 6, the expectation of $n(x,a,r_a(\mathbf{x}))$ can be bounded as follows.

$$
\begin{aligned}
\mathbb{E}[n(x,a,r_a(\mathbf{x}))] &= P(E)\mathbb{E}[n(x,a,r_a(\mathbf{x}))|E]+P(E^c)T\\
&\leq k+1,
\end{aligned}
\tag{79}
$$

in which the first step holds since even if $E$ does not hold, the number of steps in $n(x,a,r_a(\mathbf{x}))$ is no more than the total sample size $T$. The second step uses (74). From (79) and the definition of expected sample density in (24),

$$\int_{B(x,r_a(\mathbf{x}))}q_a(\mathbf{u})du\leq k+1. \tag{80}$$

It bounds the average value of $q_a$ over the neighborhood of $\mathbf{x}$. However, it does not bound $q_a(\mathbf{x})$ directly. To bound $R_a$, we introduce a new random variable $\mathbf{Z}$, with pdf

$$g(\mathbf{z}) = \frac{1}{M_Z \left[ (\eta^*(\mathbf{z}) - \eta_a(\mathbf{z})) \vee \epsilon \right]^d}, \tag{81}$$

in which $\epsilon = 4b$, with $b$ defined in (10). $M_Z$ is the constant for normalization. We then bound $R_a$ defined in (29). $R_a$ can be split into two terms:

$$
\begin{aligned}
R_a &= \int_{\mathcal{X}} (\eta^*(\mathbf{x}) - \eta_a(\mathbf{x})) q_a(\mathbf{x}) \mathbf{1}(\eta^*(\mathbf{x}) - \eta_a(\mathbf{x}) > \epsilon) d\mathbf{x} \\
&\quad + \int_{\mathcal{X}} (\eta^*(\mathbf{x}) - \eta_a(\mathbf{x})) q_a(\mathbf{x}) \mathbf{1}(\eta^*(\mathbf{x}) - \eta_a(\mathbf{x}) \le \epsilon) d\mathbf{x}.
\end{aligned}
\tag{82}
$$

To begin with, we bound the first term in (82). We show the following lemma.

**Lemma 7.** *There exists a constant $C_1$, such that*

$$\int_{\mathcal{X}} (\eta^*(\mathbf{x}) - \eta_a(\mathbf{x})) q_a(\mathbf{x}) \mathbf{1}(\eta^*(\mathbf{x}) - \eta_a(\mathbf{x}) > \epsilon) d\mathbf{x} \le C_1 M_Z \mathbb{E} \left[ \int_{B(\mathbf{Z}, r_a(\mathbf{Z}))} q_a(\mathbf{u})(\eta^*(\mathbf{u}) - \eta_a(\mathbf{u})) du \right], \tag{83}$$

*in which $\epsilon = 4b$.*

*Proof.* The proof of Lemma 7 is shown in Appendix I.4. $\square$

Now we bound the right hand side of (83). We show the following lemma.

**Lemma 8.**

$$\mathbb{E} \left[ \int_{B(\mathbf{Z}, r_a(\mathbf{Z}))} q_a(\mathbf{u})(\eta^*(\mathbf{u}) - \eta_a(\mathbf{u})) du \right] \lesssim \begin{cases} \frac{k}{M_Z c} \epsilon^{\alpha+1-d} & \text{if} \quad d > \alpha + 1 \\ \frac{k}{M_Z c} \ln \frac{1}{\epsilon} & \text{if} \quad d = \alpha + 1 \\ \frac{k}{M_Z c} & \text{if} \quad d < \alpha + 1, \end{cases} \tag{84}$$

*in which $c$ is the lower bound of pdf of contexts, which comes from Assumption 2.*

*Proof.* The proof of Lemma 8 is shown in Appendix I.5. $\square$

From Lemma 7 and 8,

$$\int_{\mathcal{X}} (\eta^*(\mathbf{x}) - \eta_a(\mathbf{x})) q_a(\mathbf{x}) \mathbf{1}(\eta^*(\mathbf{x}) - \eta_a(\mathbf{x}) > \epsilon) d\mathbf{x} \lesssim \begin{cases} k \epsilon^{\alpha+1-d} & \text{if} \quad d > \alpha + 1 \\ k \ln \frac{1}{\epsilon} & \text{if} \quad d = \alpha + 1 \\ k & \text{if} \quad d < \alpha + 1. \end{cases} \tag{85}$$

Now we bound the second term in (82). From Lemma 1, $q_a(\mathbf{x}) \le T f(\mathbf{x})$ for almost all $\mathbf{x} \in \mathcal{X}$. Thus

$$
\begin{aligned}
\int (\eta^*(\mathbf{x}) - \eta_a(\mathbf{x})) q_a(\mathbf{x}) \mathbf{1}(\eta^*(\mathbf{x}) - \eta_a(\mathbf{x}) \le \epsilon) d\mathbf{x} &\le T \int (\eta^*(\mathbf{x}) - \eta_a(\mathbf{x})) f(\mathbf{x}) \mathbf{1}(\eta^*(\mathbf{x}) - \eta_a(\mathbf{x}) \le \epsilon) d\mathbf{x} \\
&\le T \epsilon \mathrm{P}(\eta^*(\mathbf{X}) - \eta_a(\mathbf{X}) \le \epsilon) \\
&\le C_\alpha T \epsilon^{\alpha+1}.
\end{aligned}
\tag{86}
$$

Therefore, from (82), (85) and (86),

$$R_a \lesssim \begin{cases} T \epsilon^{\alpha+1} + k \epsilon^{\alpha+1-d} & \text{if} \quad d > \alpha + 1 \\ T \epsilon^{\alpha+1} + k \ln \frac{1}{\epsilon} & \text{if} \quad d = \alpha + 1 \\ T \epsilon^{\alpha+1} + k & \text{if} \quad d < \alpha + 1. \end{cases} \tag{87}$$

Recall that

$$\epsilon = 4b = 4\sqrt{\frac{2\sigma^2}{k}\ln(dT^{2d+2}|\mathcal{A}|)}. \tag{88}$$

If $d > \alpha + 1$, let $k \sim T^{\frac{2}{d+2}}$, then

$$\epsilon \sim T^{-\frac{1}{d+2}}\sqrt{\ln(dT^{2d+2}|\mathcal{A}|)}, \tag{89}$$

and

$$R_a \lesssim T^{1-\frac{\alpha+1}{d+2}}\ln^{\frac{\alpha+1}{2}}(dT^{2d+2}|\mathcal{A}|). \tag{90}$$

If $d \le \alpha + 1$, let $k \sim T^{\frac{2}{\alpha+3}}$, then

$$\epsilon \sim T^{-\frac{1}{\alpha+3}}\sqrt{\ln(dT^{2d+2}|\mathcal{A}|)}, \tag{91}$$

and

$$R_a \lesssim T^{\frac{2}{\alpha+3}}\ln^{\frac{\alpha+1}{2}}(dT^{(}2d+2)|\mathcal{A}|). \tag{92}$$

Theorem 3 can then be proved using by $R = \sum_{a\in\mathcal{A}} R_a$ stated in Lemma 2.

# F. Proof of Theorem 4

Recall the expression of regret shown in Lemma 2. We decompose $R_a$ as follows.

$$
\begin{aligned}
R_a \;=\; & \int_{\mathcal{X}} (\eta^*(\mathbf{x}) - \eta_a(\mathbf{x}))q_a(\mathbf{x})\mathbf{1}(\eta^*(\mathbf{x}) - \eta_a(\mathbf{x}) \le \epsilon)d\mathbf{x} \\
& + \int_{\mathcal{X}} (\eta^*(\mathbf{x}) - \eta_a(\mathbf{x}))q_a(\mathbf{x})\mathbf{1}\left(\eta^*(\mathbf{x}) - \eta_a(\mathbf{x}) > \epsilon, f(\mathbf{x}) > \frac{k}{T\epsilon^d}\right)d\mathbf{x} \\
& + \int_{\mathcal{X}} (\eta^*(\mathbf{x}) - \eta_a(\mathbf{x}))q_a(\mathbf{x})\mathbf{1}\left(\eta^*(\mathbf{x}) - \eta_a(\mathbf{x}) > \epsilon, \frac{k}{T} < f(\mathbf{x}) \le \frac{k}{T\epsilon^d}\right)d\mathbf{x} \\
& + \int_{\mathcal{X}} (\eta^*(\mathbf{x}) - \eta_a(\mathbf{x}))q_a(\mathbf{x})\mathbf{1}\left(\eta^*(\mathbf{x}) - \eta_a(\mathbf{x}) > \epsilon, f(\mathbf{x}) \le \frac{k}{T}\right)d\mathbf{x} \\
:=\; & I_1 + I_2 + I_3 + I_4, 
\end{aligned} \tag{93}
$$

in which $\epsilon$ is the same as the proof of Theorem 3 in Appendix 5, i.e. $\epsilon = 4b$.

**Bound of $I_1$.** From Lemma 1, $q(\mathbf{x}) \le Tf(\mathbf{x})$ for almost all $\mathbf{x} \in \mathcal{X}$. Hence

$$
\begin{aligned}
I_1 \;&\le\; T\int_{\mathcal{X}} (\eta^*(\mathbf{x}) - \eta_a(\mathbf{x}))\mathbf{1}(\eta^*(\mathbf{x}) - \eta_a(\mathbf{x}) \le \epsilon)d\mathbf{x} \\
&\le\; T\epsilon \mathrm{P}(\eta^*(\mathbf{X}) - \eta_a(\mathbf{X}) \le \epsilon) \\
&\le\; C_\alpha T\epsilon^{1+\alpha}.
\end{aligned} \tag{94}
$$

**Bound of $I_2$.** The regret of the high density region can be bounded similarly as the regret for pdf bounded away from zero. Follow the proof of Theorem 3 in Appendix 5, define

$$g(\mathbf{z}) = \frac{1}{M_Z\left[(\eta^*(\mathbf{z}) - \eta_a(\mathbf{z})) \vee \epsilon\right]^d}\mathbf{1}\left(f(\mathbf{z}) > \frac{k}{T\epsilon^d}\right). \tag{95}$$

Similar to Lemma 5,

$$I_2 \lesssim M_Z \int_{B(\mathbf{Z}, r_a(\mathbf{Z}))} q_a(\mathbf{u})(\eta^*(\mathbf{u}) - \eta_a(\mathbf{u}))du. \tag{96}$$

Similar to Lemma 6, now we replace $c$ with $k/(T\epsilon^d)$. Then

$$\int_{B(\mathbf{Z},r_a(\mathbf{Z}))} q_a(\mathbf{u})(\eta^*(\mathbf{u}) - \eta_a(\mathbf{u}))du \lesssim \begin{cases} \frac{T\epsilon^d}{M_Z}\epsilon^{\alpha+1-d} & \text{if} \quad d > \alpha+1 \\ \frac{T\epsilon^d}{M_Z}\ln\frac{1}{\epsilon} & \text{if} \quad d = \alpha+1 \\ \frac{T\epsilon^d}{M_Z} & \text{if} \quad d < \alpha+1. \end{cases} \tag{97}$$

Therefore

$$I_2 \lesssim \begin{cases} T\epsilon^d\epsilon^{\alpha+1-d} & \text{if} \quad d > \alpha+1 \\ T\epsilon^d\ln\frac{1}{\epsilon} & \text{if} \quad d = \alpha+1 \\ T\epsilon^d & \text{if} \quad d < \alpha+1. \end{cases} \tag{98}$$

**Bound of $I_3$.** Here we introduce the following lemma.

**Lemma 9.** *(Restated from Lemma 6 in (Zhao & Lai, 2021b))* For any $0 < a < b$,

$$\mathbb{E}[f^{-p}(\mathbf{X})\mathbf{1}(a \le f(\mathbf{X}) < b)] \lesssim \begin{cases} b^{\beta-p} & \text{if} \quad p > \beta \\ \ln\frac{b}{a} & \text{if} \quad p = \beta \\ a^{\beta-p} & \text{if} \quad p < \beta. \end{cases} \tag{99}$$

*Proof.* The proof of Lemma 9 can follow that of Lemma 6 in (Zhao & Lai, 2021b). For completeness, we show the proof in Appendix I.9. □

Based on Lemma 9, $I_3$ can be bounded by

$$\begin{aligned} I_3 &= \int_{\mathcal{X}} (\eta^*(\mathbf{x}) - \eta_a(\mathbf{x}))q_a(\mathbf{x})\mathbf{1}\left(\eta^*(\mathbf{x}) - \eta_a(\mathbf{x}) > \epsilon, \frac{k}{T} < f(\mathbf{x}) \le \frac{k}{\epsilon^d}\right) d\mathbf{x} \\ &\lesssim \int_{\mathcal{X}} \left(\frac{k}{Tf(\mathbf{x})}\right)^{\frac{1}{d}} Tf(\mathbf{x})\mathbf{1}\left(\eta^*(\mathbf{x}) - \eta_a(\mathbf{x}) > \epsilon, \frac{k}{T} < f(\mathbf{x}) \le \frac{k}{T\epsilon^d}\right) d\mathbf{x} \\ &\le T\left(\frac{k}{T}\right)^{\frac{1}{d}} \int \mathbb{E}\left[f^{-\frac{1}{d}}(\mathbf{X})\mathbf{1}\left(\frac{k}{T} < f(\mathbf{x}) < \frac{k}{T\epsilon^d}\right)\right] \\ &\lesssim \begin{cases} T\left(\frac{k}{T}\right)^{\beta} & \text{if} \quad \beta < \frac{1}{d} \\ T\left(\frac{k}{T}\right)^{\frac{1}{d}}\ln\frac{1}{\epsilon} & \text{if} \quad \beta = \frac{1}{d} \\ T\left(\frac{k}{T}\right)^{\beta}\epsilon^{1-d\beta} & \text{if} \quad \beta > \frac{1}{d}. \end{cases} \end{aligned}$$

**Bound of $I_4$.**

$$\begin{aligned} I_4 &\le TM\int f(\mathbf{x})\mathbf{1}\left(f(\mathbf{x}) \le \frac{k}{T}\right) d\mathbf{x} \\ &\le TM\mathrm{P}(f(\mathbf{X}) \le \frac{k}{T}) \\ &\lesssim T\left(\frac{k}{T}\right)^{\beta}. \end{aligned} \tag{100}$$

Now we bound $R_a$ by selecting $k$ to minimize the sum of $I_1, I_2, I_3, I_4$. Recall that $\epsilon = 4b$, in which $b$ is defined in (10), thus $\epsilon \sim \sqrt{\ln(dT^{2d+2}|\mathcal{A}|)/k}$.

(1) If $d > \alpha + 1$ and $\beta > 1/d$, then with $k \sim T^{\frac{2\beta}{\alpha+(d+2)\beta}}$,

$$\begin{aligned} R &\lesssim T\left(\epsilon^{1+\alpha} + \left(\frac{k}{T}\right)^{\beta}\epsilon^{1-d\beta}\right) \\ &\sim T^{1-\frac{\beta(\alpha+1)}{\alpha+(d+2)\beta}}\ln^{\frac{\alpha+1}{2}}(dT^{2d+2}|\mathcal{A}|). \end{aligned} \tag{101}$$

(2) If $d \leq \alpha + 1$ and $\beta > 1/d$, then with $k \sim T^{\frac{2\beta}{d-1+\beta(d+2)}}$,

$$
\begin{aligned}
R &\lesssim T\left(\epsilon^d + \left(\frac{k}{T}\right)^\beta \epsilon^{1-d\beta}\right) \\
&\sim T^{1-\frac{\beta d}{d-1+(d+2)\beta}} \ln^{\frac{d}{2}}(dT^{2d+2}|\mathcal{A}|).
\end{aligned}
\tag{102}
$$

(3) If $d > \alpha + 1$ and $\beta \leq 1/d$, then with $k \sim T^{\frac{2\beta}{1+\alpha+2\beta}}$,

$$
\begin{aligned}
R &\lesssim T\epsilon^{1+\alpha} + T\left(\frac{k}{N}\right)^\beta \\
&\sim T^{1-\frac{\beta(\alpha+1)}{1+\alpha+2\beta}} \ln^{\frac{\alpha+1}{2}}(dT^{2d+2}|\mathcal{A}|).
\end{aligned}
\tag{103}
$$

(4) If $d \leq \alpha + 1$ and $\beta \leq 1/d$, then with $k \sim T^{\frac{2\beta}{d+2\beta}}$,

$$
\begin{aligned}
R &\lesssim T\epsilon^d + T\left(\frac{k}{N}\right)^\beta \\
&\sim T^{1-\frac{\beta d}{d+2\beta}} \ln^{\frac{d}{2}}(dT^{2d+2}|\mathcal{A}|).
\end{aligned}
\tag{104}
$$

Combine all these cases, we conclude that

$$
R \lesssim T^{1-\frac{\beta \min(d,\alpha+1)}{\min(d-1,\alpha)+\max(1,d\beta)+2\beta}} |\mathcal{A}| \ln^{\frac{1}{2}\max(d,\alpha+1)}(dT^{2d+2}|\mathcal{A}|).
\tag{105}
$$

The proof of Theorem 4 is complete.

## G. Proof of Theorem 5

To begin with, similar to Lemma 4 and Lemma 5, we show the following lemmas.

**Lemma 10.**

$$
P\left(\sup_{x,a,k}\left|\frac{1}{\sqrt{k}}\sum_{i \in \mathcal{N}_{t,k}(\mathbf{x},a)} W_i\right| > u\right) \leq dT^{2d+1}|\mathcal{A}|e^{-\frac{u^2}{2\sigma^2}},
\tag{106}
$$

with $\mathcal{N}_{t,k}(s,a)$ being the set of $k$ neighbors among $\{\mathbf{X}_i | i < t, A_i = a\}$.

*Proof.* From Lemma 4,

$$
P\left(\sup_{x,a}\left|\frac{1}{\sqrt{k}}\sum_{i \in \mathcal{N}_{t,k}(\mathbf{x},a)} W_i\right| > u\right) \leq dT^{2d}|\mathcal{A}|e^{-\frac{u^2}{2\sigma^2}}.
\tag{107}
$$

Lemma 10 can then be proved by taking a union bound over all $k$. $\qquad\square$

**Lemma 11.** *Define event E, such that $E = 1$ if*

$$
\left|\frac{1}{\sqrt{k}}\sum_{i \in \mathcal{N}_{t,k}(\mathbf{x},a)} W_i\right| \leq \sqrt{2\sigma^2 \ln(dT^{2d+3}|\mathcal{A}|)}
\tag{108}
$$

*for all $x, a, k, t$, then $P(E) \geq 1 - 1/T$. Moreover, under E,*

$$
\eta_a(\mathbf{x}) \leq \hat{\eta}_{a,t}(\mathbf{x}) \leq \eta_a(\mathbf{x}) + 2b_{a,t}(\mathbf{x}) + 2L\rho_{a,t}(\mathbf{x}).
\tag{109}
$$

*Proof.* The proof is similar to the proof of Lemma 5.

$$|\hat{\eta}_{a,t}(\mathbf{x}) - (\eta_a(\mathbf{x}) + b_{a,t}(\mathbf{x}) + L\rho_{a,t}(\mathbf{x}))|$$

$$\leq \left| \frac{1}{k_{a,t}(\mathbf{x})} \sum_{i \in \mathcal{N}_t(\mathbf{x},a)} (Y_i - \eta_a(\mathbf{x})) \right|$$

$$\leq \left| \frac{1}{k_{a,t}(\mathbf{x})} \sum_{i \in \mathcal{N}_t(\mathbf{x},a)} (Y_i - \eta_a(\mathbf{X}_i)) \right| + \frac{1}{k_{a,t}(\mathbf{x})} \sum_{i \in \mathcal{N}_t(\mathbf{x},a)} |\eta_a(\mathbf{X}_i) - \eta_a(\mathbf{x})|$$

$$\leq L\rho_{a,t}(\mathbf{x}) + b_{a,t}(\mathbf{x}). \tag{110}$$

$\square$

With these preparations, we then bound the number of steps around each $\mathbf{x}$ in the next lemma, which is crucially different with Lemma 6. Here we keep the definition $n(x,a,r) := \sum_{t=1}^{T} \mathbf{1}\left(\|\mathbf{X}_t - \mathbf{x}\| < r, A_t = a\right)$ to be the same as (76), but change the definition of $r_a$ and $n_a$ as follows.

**Lemma 12.** *Define*

$$r_a(\mathbf{x}) = \frac{1}{2L\sqrt{C_1}}(\eta^*(\mathbf{x}) - \eta_a(\mathbf{x})), \tag{111}$$

*and*

$$n_a(\mathbf{x}) = \frac{C_1 \ln T}{(\eta^*(\mathbf{x}) - \eta_a(\mathbf{x}))^2}, \tag{112}$$

*in which*

$$C_1 = \max\{4, 32\sigma^2(2d + 3 + \ln(d|\mathcal{A}|))\}. \tag{113}$$

*Then under E,*

$$n(x, a, r_a(\mathbf{x})) \leq n_a(\mathbf{x}). \tag{114}$$

*Proof.* The proof of Lemma 12 is shown in Appendix I.6. $\square$

From Lemma 12,

$$\mathbb{E}[n(x, a, r_a(\mathbf{x}))] \leq \mathrm{P}(E)\mathbb{E}[n(x, a, r_a(\mathbf{x}))|E] + \mathrm{P}(E^c)\mathbb{E}[n(x, a, r_a(\mathbf{x}))|E^c]$$
$$\leq n_a(\mathbf{x}) + 1. \tag{115}$$

From the definition of $q_a$ in (24),

$$\int_{B(x,r_a(\mathbf{x}))} q_a(\mathbf{u})du \leq n_a(\mathbf{x}) + 1. \tag{116}$$

Now we bound $R_a$. Similar to (81), let random variable $\mathbf{Z}$ follows a distribution with pdf $g$:

$$g(\mathbf{z}) = \frac{1}{M_Z[(\eta^*(\mathbf{z}) - \eta_a(\mathbf{z})) \vee \epsilon]^d}. \tag{117}$$

The difference with the case with fixed $k$ is that in (81), $\epsilon = 4b$. However, now $b_{a,t}(\mathbf{x})$ varies among $\mathbf{x}$, thus we do not determine $\epsilon$ based on $b$. Instead, for the adaptive nearest neighbor method, $\epsilon$ will be determined after we get the final bound of $R_a$.

We show the following lemma.

**Lemma 13.** *There exists a constant $C_2$, such that*

$$\int_{\mathcal{X}} (\eta^*(\mathbf{x}) - \eta_a(\mathbf{x}))q_a(\mathbf{x})\mathbf{1}(\eta^*(\mathbf{x}) - \eta_a(\mathbf{x}) > \epsilon)d\mathbf{x} \le C_2 M_Z \mathbb{E}\left[\int_{B(\mathbf{Z}, r_a(\mathbf{Z}))} q_a(\mathbf{u})(\eta^*(\mathbf{u}) - \eta_a(\mathbf{u}))du\right]. \qquad (118)$$

*Proof.* The proof of Lemma 13 is shown in Appendix I.7. $\qquad\square$

We then bound the right hand side of (118).

**Lemma 14.**

$$\mathbb{E}\left[\int_{B(\mathbf{Z}, r_a(\mathbf{Z}))} q_a(\mathbf{u})(\eta^*(\mathbf{u}) - \eta_a(\mathbf{u}))du\right] \lesssim \frac{1}{M_Z}\left(\epsilon^{\alpha-d-1}\ln T + T\epsilon^{1+\alpha}\right). \qquad (119)$$

*Proof.* The proof of Lemma 14 is shown in Appendix I.8. $\qquad\square$

From Lemma 13 and 14,

$$\int_{\mathcal{X}} (\eta^*(\mathbf{x}) - \eta_a(\mathbf{x}))q_a(\mathbf{x})\mathbf{1}(\eta^*(\mathbf{x}) - \eta_a(\mathbf{x}) > \epsilon)d\mathbf{x} \lesssim \epsilon^{\alpha-d-1}\ln T + T\epsilon^{1+\alpha}. \qquad (120)$$

Moreover,

$$\begin{aligned}
\int_{\mathcal{X}} (\eta^*(\mathbf{x}) - \eta_a(\mathbf{x}))q_a(\mathbf{x})\mathbf{1}(\eta^*(\mathbf{x}) - \eta_a(\mathbf{x}) \le \epsilon)d\mathbf{x} &\le T\epsilon\int f(\mathbf{x})\mathbf{1}(\eta^*(\mathbf{x}) - \eta_a(\mathbf{x}) \le \epsilon)d\mathbf{x} \\
&\lesssim T\epsilon^{1+\alpha}.
\end{aligned} \qquad (121)$$

Therefore

$$R_a \lesssim \epsilon^{\alpha-d-1}\ln T + T\epsilon^{1+\alpha}. \qquad (122)$$

Note that we have not specified the value of $\epsilon$ earlier. Therefore, in (122), $\epsilon$ can take any values. We can then select $\epsilon$ to minimize the right hand side of (122). Therefore, let

$$\epsilon \sim \left(\frac{\ln T}{T}\right)^{\frac{1}{d+2}}, \qquad (123)$$

then

$$R_a \lesssim T\left(\frac{T}{\ln T}\right)^{-\frac{1+\alpha}{d+2}}. \qquad (124)$$

The overall regret can then be bounded by summation over $a$:

$$R \lesssim T|\mathcal{A}|\left(\frac{T}{\ln T}\right)^{-\frac{1+\alpha}{d+2}}. \qquad (125)$$

# H. Proof of Theorem 6

Define

$$g(\mathbf{z}) = \frac{1}{M_Z(\eta^*(\mathbf{z}) - \eta_a(\mathbf{z}))^d}\mathbf{1}\left(f(\mathbf{z}) \ge \frac{1}{T}, \eta^*(\mathbf{z}) - \eta_a(\mathbf{z}) > \epsilon(\mathbf{x})\right), \qquad (126)$$

in which

$$\epsilon(\mathbf{x}) = (Tf(\mathbf{x}))^{-\frac{1}{d+2}}, \qquad (127)$$

and $M_Z$ is the normalization constant, which ensures that $\int g(\mathbf{z})dz = 1$. Let $\mathbf{Z}$ be a random variable with pdf $g$. We then bound $R_a$ for the case with unbounded support on the contexts, under Assumption 2 and 3.

$$
\begin{aligned}
R_a &= \int_{\mathcal{X}} (\eta^*(\mathbf{x}) - \eta_a(\mathbf{x}))q_a(\mathbf{x})d\mathbf{x} \\
&= \int_{\mathcal{X}} (\eta^*(\mathbf{x}) - \eta_a(\mathbf{x}))q_a(\mathbf{x})\mathbf{1}\left(\eta^*(\mathbf{x}) - \eta_a(\mathbf{x}) > \epsilon(\mathbf{x}), f(\mathbf{x}) \geq \frac{1}{T}\right)d\mathbf{x} \\
&\quad + \int_{\mathcal{X}} (\eta^*(\mathbf{x}) - \eta_a(\mathbf{x}))q_a(\mathbf{x})\mathbf{1}\left(\eta^*(\mathbf{x}) - \eta_a(\mathbf{x}) \leq \epsilon(\mathbf{x}), f(\mathbf{x}) \geq \frac{1}{T}\right)d\mathbf{x} \\
&\quad + \int_{\mathcal{X}} (\eta^*(\mathbf{x}) - \eta_a(\mathbf{x}))q_a(\mathbf{x})\mathbf{1}\left(f(\mathbf{x}) < \frac{1}{T}\right)d\mathbf{x} \\
&:= I_1 + I_2 + I_3.
\end{aligned}
\tag{128}
$$

Now we bound three terms in (128) separately.

**Bound of $I_1$.** Following Lemma 7 and 13, it can be shown that for some constant $C_3$, such that

$$
\int_{\mathcal{X}} (\eta^*(\mathbf{x}) - \eta_a(\mathbf{x}))q_a(\mathbf{x})\mathbf{1}(\eta^*(\mathbf{x}) - \eta_a(\mathbf{x}) > \epsilon(\mathbf{x}))d\mathbf{x} \leq C_3 M_Z \mathbb{E}\left[\int_{B(\mathbf{Z}, r_a(\mathbf{Z}))} q_a(\mathbf{u})(\eta^*(\mathbf{u}) - \eta_a(\mathbf{u}))du\right].
\tag{129}
$$

The right hand side of (129) can be bounded as follows.

$$
\begin{aligned}
&\mathbb{E}\left[\int_{B(\mathbf{Z}, r_a(\mathbf{Z}))} q_a(\mathbf{u})(\eta^*(\mathbf{u}) - \eta_a(\mathbf{u}))du\right] \\
&\leq \frac{3}{2}\mathbb{E}\left[\int_{B(\mathbf{Z}, r_a(\mathbf{Z}))} q_a(\mathbf{u})(\eta^*(\mathbf{Z}) - \eta_a(\mathbf{Z}))du\right] \\
&\leq \frac{3}{2}\mathbb{E}\left[(n_a(\mathbf{Z}) + 1)(\eta^*(\mathbf{Z}) - \eta_a(\mathbf{Z}))\right] \\
&\leq \frac{3}{2M_Z}\int (n_a(\mathbf{z}) + 1)(\eta^*(\mathbf{z}) - \eta_a(\mathbf{z}))g(\mathbf{z})dz \\
&\leq \frac{3}{2M_Z}\int \left(\frac{C_1 \ln T}{\eta^*(\mathbf{z}) - \eta_a(\mathbf{z})} + \eta^*(\mathbf{z}) - \eta_a(\mathbf{z})\right)\frac{1}{(\eta^*(\mathbf{z}) - \eta_a(\mathbf{z}))^d}\mathbf{1}\left(\eta(\mathbf{z}) \geq \epsilon(\mathbf{z}), f(\mathbf{z}) \geq \frac{1}{T}\right)dz \\
&\lesssim \frac{\ln T}{M_Z}\int (\eta^*(\mathbf{z}) - \eta_a(\mathbf{z}))^{-(d+1)}\mathbf{1}\left(\eta(\mathbf{z}) \geq \epsilon(\mathbf{z}), f(\mathbf{z}) \geq \frac{1}{T}\right)dz.
\end{aligned}
\tag{130}
$$

Hence

$$
I_1 \lesssim \ln T \int (\eta^*(\mathbf{x}) - \eta_a(\mathbf{x}))^{-(d+1)}\mathbf{1}\left(\eta(\mathbf{x}) \geq \epsilon(\mathbf{x}), f(\mathbf{x}) \geq \frac{1}{T}\right)d\mathbf{x}.
\tag{131}
$$

Let $c \in R$ that will be determined later.

$$
\begin{aligned}
&\int (\eta^*(\mathbf{x}) - \eta_a(\mathbf{x}))^{-(d+1)}\mathbf{1}(\eta^*(\mathbf{x}) - \eta_a(\mathbf{x}) \geq \epsilon(\mathbf{x}), f(\mathbf{x}) \geq c)d\mathbf{x} \\
&\leq \frac{1}{c}\int (\eta^*(\mathbf{x}) - \eta_a(\mathbf{x}))^{-(d+1)}\mathbf{1}\left(\eta^*(\mathbf{x}) - \eta_a(\mathbf{x}) \geq (Tc)^{-\frac{1}{d+2}}\right)f(\mathbf{x})d\mathbf{x} \\
&= \frac{1}{c}\mathbb{E}\left[(\eta^*(\mathbf{X}) - \eta_a(\mathbf{X}))^{-(d+1)}\mathbf{1}\left(\eta^*(\mathbf{X}) - \eta_a(\mathbf{X}) \geq (Tc)^{-\frac{1}{d+2}}\right)\right] \\
&\leq \frac{1}{c}(Tc)^{\frac{d+1}{d+2}\left(1 - \frac{\alpha}{d+1}\right)} \\
&= T(Tc)^{-\frac{\alpha+1}{d+2}}.
\end{aligned}
\tag{132}
$$

To bound the integration of the other side, i.e. $1/T \leq f(\mathbf{x}) < c$, we use Lemma 9.

$$\int (\eta^*(\mathbf{x}) - \eta_a(\mathbf{x}))^{-(d+1)} \mathbf{1}\left(\eta(\mathbf{x}) \geq \epsilon(\mathbf{x}), \frac{1}{T} \leq f(\mathbf{x}) < c\right) d\mathbf{x}$$

$$\leq \int \epsilon^{-(d+1)}(\mathbf{x}) \mathbf{1}\left(\frac{1}{T} \leq f(\mathbf{x}) < c\right) d\mathbf{x}$$

$$= T^{\frac{d+1}{d+2}} \int f^{\frac{d+1}{d+2}}(\mathbf{x}) \mathbf{1}\left(\frac{1}{T} \leq f(\mathbf{x}) < c\right) d\mathbf{x}$$

$$= T^{\frac{d+1}{d+2}} \mathbb{E}\left[f^{-\frac{1}{d+2}}(\mathbf{X}) \mathbf{1}\left(\frac{1}{T} \leq f(\mathbf{X}) < c\right)\right]$$

$$\lesssim \begin{cases} T^{\frac{d+1}{d+2}}\left(T^{\frac{1}{d+2}-\beta} + c^{\beta - \frac{1}{d+2}}\right) & \text{if} \quad \beta \neq \frac{1}{d+2} \\ T^{\frac{d+1}{d+2}} \ln(Tc) & \text{if} \quad \beta = \frac{1}{d+2}. \end{cases} \tag{133}$$

From (132) and (133),

$$I_1 \lesssim \begin{cases} T \ln T \left[(Tc)^{-\frac{\alpha+1}{d+2}} + T^{-\frac{1}{d+2}} c^{\beta - \frac{1}{d+2}} + T^{-\beta}\right] & \text{if} \quad \beta \neq \frac{1}{d+2} \\ T \ln T \left[(Tc)^{-\frac{\alpha+1}{d+2}} + T^{-\frac{1}{d+2}} \ln(Tc)\right] & \text{if} \quad \beta = \frac{1}{d+2}. \end{cases} \tag{134}$$

To minimize (134), let

$$c = T^{-\frac{\alpha}{\alpha+(d+2)\beta}}, \tag{135}$$

then

$$I_1 \lesssim \begin{cases} T^{1-\frac{(\alpha+1)\beta}{\alpha+(d+2)\beta}} \ln T + T^{1-\beta} \ln T & \text{if} \quad \beta \neq \frac{1}{d+2} \\ T^{\frac{d+1}{d+2}} \ln^2 T & \text{if} \quad \beta = \frac{1}{d+2}. \end{cases} \tag{136}$$

**Bound of $I_2$.** We still discuss $f(\mathbf{x}) \geq c$ and $1/T \leq f(\mathbf{x}) < c$ separately. For $f(\mathbf{x}) \geq c$,

$$\int (\eta^*(\mathbf{x}) - \eta_a(\mathbf{x})) q_a(\mathbf{x}) \mathbf{1}(\eta^*(\mathbf{x}) - \eta_a(\mathbf{x}) \leq \epsilon(\mathbf{x}), f(\mathbf{x}) \geq c) d\mathbf{x}$$

$$\leq T \int (\eta^*(\mathbf{x}) - \eta_a(\mathbf{x})) \mathbf{1}\left(\eta^*(\mathbf{x}) - \eta_a(\mathbf{x}) \leq (Tc)^{-\frac{1}{d+2}}\right) f(\mathbf{x}) d\mathbf{x}$$

$$\leq T(Tc)^{-\frac{1}{d+2}} \mathrm{P}\left(\eta^*(\mathbf{X}) - \eta_a(\mathbf{X}) \leq (Tc)^{-\frac{1}{d+2}}\right)$$

$$\lesssim T(Tc)^{-\frac{1+\alpha}{d+2}}. \tag{137}$$

For $1/T \leq f(\mathbf{x}) < c$,

$$\int (\eta^*(\mathbf{x}) - \eta_a(\mathbf{x})) q_a(\mathbf{x}) \mathbf{1}\left(\eta^*(\mathbf{x}) - \eta_a(\mathbf{x}) \leq \epsilon(\mathbf{x}), \frac{1}{T} \leq f(\mathbf{x}) < c\right) d\mathbf{x}$$

$$\leq T \int \epsilon(\mathbf{x}) f(\mathbf{x}) \mathbf{1}\left(\frac{1}{T} \leq f(\mathbf{x}) < c\right) d\mathbf{x}$$

$$= T^{\frac{d+1}{d+2}} \int f^{\frac{d+1}{d+2}}(\mathbf{x}) \mathbf{1}\left(\frac{1}{T} \leq f(\mathbf{x}) < c\right) d\mathbf{x}$$

$$\lesssim \begin{cases} T^{\frac{d+1}{d+2}}(T^{\frac{1}{d+2}-\beta} + c^{\beta-\frac{1}{d+2}}) & \text{if} \quad \beta \neq \frac{1}{d+2} \\ T^{\frac{d+1}{d+2}} \ln(Tc) & \text{if} \quad \beta = \frac{1}{d+2}. \end{cases} \tag{138}$$

Similar to $I_1$, pick $c = T^{-\alpha/(\alpha+(d+2)\beta)}$.

**Bound of $I_3$.** From Assumption 3(b), $\eta^*(\mathbf{x}) - \eta_a(\mathbf{x}) \leq M$. Moreover, from Lemma 1, $q(\mathbf{x}) \leq T f(\mathbf{x})$ for almost all $\mathbf{x} \in \mathcal{X}$. Hence

$$\int (\eta^*(\mathbf{x}) - \eta_a(\mathbf{x})) q_a(\mathbf{x}) \mathbf{1}\left(f(\mathbf{x}) < \frac{1}{T}\right) d\mathbf{x} \leq MT \int f(\mathbf{x}) \mathbf{1}\left(f(\mathbf{x}) < \frac{1}{T}\right) d\mathbf{x} \lesssim T^{1-\beta}. \tag{139}$$

Combine $I_1$, $I_2$ and $I_3$,

$$R_a \lesssim \begin{cases} T^{1 - \frac{(\alpha+1)\beta}{\alpha+(d+2)\beta}} \ln T + T^{1-\beta} \ln T & \text{if} \quad \beta \neq \frac{1}{d+2} \\ T^{\frac{d+1}{d+2}} \ln^2 T & \text{if} \quad \beta = \frac{1}{d+2}. \end{cases} \tag{140}$$

Theorem 6 can then proved by $R = \sum_{a \in \mathcal{A}} R_a$.

# I. Proof of Lemmas

## I.1. Proof of Lemma 4

From Assumption 2(c), $W_i$ is subgaussian with parameter $\sigma^2$. Therefore for any fixed set $I \subset \{1, \dots, T\}$ with $|I| = k$,

$$\mathbb{E}\left[\exp\left(\lambda \sum_{i \in I} W_i\right)\right] \leq \exp\left(\frac{k}{2} \lambda^2 \sigma^2\right), \tag{141}$$

and

$$\begin{aligned} \mathrm{P}\left(\frac{1}{k} \sum_{i \in I} W_i > u\right) &\leq \inf_\lambda e^{-\lambda u} \mathbb{E}\left[\exp\left(\frac{\lambda}{k} \sum_{i \in I} W_i\right)\right] \\ &\leq \inf_\lambda e^{-\lambda u} \exp\left(\frac{\lambda^2 \sigma^2}{2k}\right) \\ &= \exp\left(-\frac{ku^2}{2\sigma^2}\right). \end{aligned} \tag{142}$$

Now we need to give a union bound[3]. Let $A_{ij}$ be $d-1$ dimensional hyperplane that bisects $\mathbf{X}_i$, $\mathbf{X}_j$. Then the number of planes is at most $N_p = T(T-1)/2$. Note that $N_p$ planes divide a $d$ dimensional space into at most $N_r = \sum_{j=0}^d \binom{N_p}{j}$ regions. Therefore

$$N_r \leq \sum_{j=0}^d \binom{\frac{1}{2}T(T-1)}{j} \leq d\left(\frac{1}{2}T(T-1)\right)^d < dT^{2d}. \tag{143}$$

The $k$ nearest neighbors for all $\mathbf{x}$ within a region should be the same. Combining with the action space $\mathcal{A}$, there are at most $N_r|\mathcal{A}|$ regions. Hence

$$|\{\mathcal{N}_t(\mathbf{x}, a) | \mathbf{x} \in \mathcal{X}, a \in \mathcal{A}\}| \leq dT^{2d}|\mathcal{A}|. \tag{144}$$

Therefore

$$\mathrm{P}\left(\cup_{\mathbf{x} \in \mathcal{X}} \cup_{a \in \mathcal{A}} \left\{\frac{1}{k}\left|\sum_{i \in \mathcal{N}_t(\mathbf{x}, a)} W_i\right| > u\right\}\right) \leq dT^{2d}|\mathcal{A}|e^{-\frac{ku^2}{2\sigma^2}}. \tag{145}$$

The proof is complete.

---

[3]The construction of hyperplanes follows the proof of Lemma 3 in (Jiang, 2019) and Appendix H.5 in (Zhao & Wan, 2024).

### I.2. Proof of Lemma 5

From (9), with $|\{i < t | A_i = a\}| \geq k$,

$$
\begin{aligned}
\hat{\eta}_{a,t}(\mathbf{x}) &= \frac{1}{k} \sum_{i \in \mathcal{N}_t(\mathbf{x},a)} Y_i + b + L\rho_{a,t}(\mathbf{x}) \\
&= \frac{1}{k} \sum_{i \in \mathcal{N}_t(\mathbf{x},a)} \eta_a(\mathbf{X}_i) + \frac{1}{k} \sum_{i \in \mathcal{N}_t(\mathbf{x},a)} W_i + b + L\rho_{a,t}(\mathbf{x}).
\end{aligned}
\tag{146}
$$

Hence

$$
\begin{aligned}
|\hat{\eta}_{a,t}(\mathbf{x}) - (\eta_a(\mathbf{x}) + b + L\rho_{a,t}(\mathbf{x}))| &\leq \frac{1}{k} \sum_{i \in \mathcal{N}_t(\mathbf{x},a)} |\eta_a(\mathbf{X}_i) - \eta_a(\mathbf{x})| + \left| \frac{1}{k} \sum_{i \in \mathcal{N}_t(\mathbf{x},a)} W_i \right| \\
&\leq L\rho_{a,t}(\mathbf{x}) + b,
\end{aligned}
\tag{147}
$$

which comes from Assumption 2(d) and Lemma 4. Lemma 5 can then be proved using (147).

### I.3. Proof of Lemma 6

We prove Lemma 6 by contradiction. If $n(x, a, r_a(\mathbf{x})) > k$, then let

$$
t = \max\{\tau | \|\mathbf{X}_\tau - \mathbf{x}\| \leq r_a(\mathbf{x}), A_\tau = a\}
\tag{148}
$$

be the last step falling in $B(x, r_a(\mathbf{x}))$ with action a. Then $B(x, r_a(\mathbf{x})) \leq B(\mathbf{X}_t, 2r_a(\mathbf{x}))$, and thus there are at least $k$ points in $B(\mathbf{X}_t, 2r_a(\mathbf{x}))$. Therefore,

$$
\rho_{a,t}(\mathbf{x}) < 2r_a(\mathbf{x})
\tag{149}
$$

Denote

$$
a^*(\mathbf{x}) = \arg\max_a \eta_a(\mathbf{x})
\tag{150}
$$

as the best action at context $\mathbf{x}$. $A_t = a$ is selected only if the UCB of action $a$ is not less than the UCB of action $a^*(\mathbf{x})$, i.e.

$$
\hat{\eta}_{a,t}(\mathbf{X}_t) \geq \hat{\eta}_{a^*(\mathbf{x}),t}(\mathbf{X}_t).
\tag{151}
$$

From Lemma 5,

$$
\hat{\eta}_{a,t}(\mathbf{X}_t) \leq \eta_a(\mathbf{X}_t) + 2b + 2L\rho_{a,t}(\mathbf{X}_t),
\tag{152}
$$

and

$$
\hat{\eta}_{a^*(\mathbf{x}),t}(\mathbf{X}_t) \geq \eta_{a^*(\mathbf{x})}(\mathbf{X}_t) = \eta^*(\mathbf{X}_t).
\tag{153}
$$

From (151), (152) and (153),

$$
\eta_a(\mathbf{X}_t) + 2b + 2L\rho_{a,t}(\mathbf{X}_t) \geq \eta^*(\mathbf{X}_t),
\tag{154}
$$

which yields

$$
\begin{aligned}
\rho_{a,t}(\mathbf{X}_t) &\geq \frac{\eta^*(\mathbf{X}_t) - \eta_a(\mathbf{X}_t) - 2b}{2L} \\
&\geq \frac{\eta^*(\mathbf{x}) - \eta_a(\mathbf{x}) - 2b - 2Lr_a(\mathbf{x})}{2L} \\
&= 2r_a(\mathbf{x}),
\end{aligned}
\tag{155}
$$

in which the last step comes from the definition of $r_a$ in (77). Note that (155) contradicts (149). Therefore $n(x, a, r_a(\mathbf{x})) \leq k$. The proof of Lemma 6 is complete.

## I.4. Proof of Lemma 7

$$
\mathbb{E}\left[\int_{B(\mathbf{Z}, r_a(\mathbf{Z}))} q_a(\mathbf{u})(\eta^*(\mathbf{u}) - \eta_a(\mathbf{u})) du\right]
$$

$$
= \int_{\mathcal{X}} g(\mathbf{z}) \left[\int_{B(z, r_a(\mathbf{z}))} q_a(\mathbf{u})(\eta^*(\mathbf{u}) - \eta_a(\mathbf{u})) du\right] dz
$$

$$
\overset{(a)}{\geq} \int_{\mathcal{X}} \int_{B(u, \frac{3}{4} r_a(\mathbf{u}))} g(\mathbf{z}) q_a(\mathbf{u})(\eta^*(\mathbf{u}) - \eta_a(\mathbf{u})) dz du
$$

$$
\geq \int_{\mathcal{X}} \left[\inf_{\|z-u\| \leq 3r_a(\mathbf{u})/4} g(\mathbf{z}) \cdot \left(\frac{3}{4}\right)^d r_a^d(\mathbf{u})\right] q_a(\mathbf{u})(\eta^*(\mathbf{u}) - \eta_a(\mathbf{u})) dz du
$$

$$
\overset{(b)}{\geq} \left(\frac{3}{4}\right)^d \left(\frac{4}{5}\right)^d \int_{\mathcal{X}} g(\mathbf{u}) r_a^d(\mathbf{u}) q_a(\mathbf{u})(\eta^*(\mathbf{u}) - \eta_a(\mathbf{u})) du
$$

$$
\overset{(c)}{=} \left(\frac{3}{5}\right)^d \frac{1}{M_Z} \int_{\mathcal{X}} \frac{1}{[(\eta^*(\mathbf{u}) - \eta_a(\mathbf{u})) \vee \epsilon]^d} \left(\frac{\eta^*(\mathbf{u}) - \eta_a(\mathbf{u}) - 2b}{6L}\right)^d q_a(\mathbf{u})(\eta^*(\mathbf{u}) - \eta_a(\mathbf{u})) du
$$

$$
\overset{(d)}{\geq} \left(\frac{3}{5}\right)^d \frac{1}{M_Z} \int_{\mathcal{X}} \mathbf{1}(\eta^*(\mathbf{u}) - \eta_a(\mathbf{u}) > \epsilon) \frac{1}{(\eta^*(\mathbf{u}) - \eta_a(\mathbf{u}))^d} \left(\frac{\eta^*(\mathbf{u}) - \eta_a(\mathbf{u})}{12L}\right)^d q_a(\mathbf{u})(\eta^*(\mathbf{u}) - \eta_a(\mathbf{u})) du
$$

$$
= \left(\frac{3}{5}\right)^d \frac{1}{(12L)^d M_Z} \int_{\mathcal{X}} q_a(\mathbf{u})(\eta^*(\mathbf{u}) - \eta_a(\mathbf{u})) \mathbf{1}(\eta^*(\mathbf{u}) - \eta_a(\mathbf{u}) > \epsilon) du. \tag{156}
$$

Based on (156), Lemma 7 holds with

$$
C_1 = \left(\frac{5}{3}\right)^d (12L)^d. \tag{157}
$$

Now we explain some key steps in (156).

In (a), the order of integration is swapped. Note that if $u \in B(z, r_a(\mathbf{z}))$, then $\|u - z\| \leq r_a(\mathbf{z})$. From Assumption 2(d), $|\eta_a(\mathbf{u}) - \eta_a(\mathbf{z})| \leq L r_a(\mathbf{z})$. Then from (77),

$$
\begin{aligned}
r_a(\mathbf{u}) &= \frac{\eta^*(\mathbf{u}) - \eta_a(\mathbf{u}) - 2b}{6L} \\
&\leq \frac{\eta^*(\mathbf{z}) - \eta_a(\mathbf{z}) + 2L r_a(\mathbf{z}) - 2b}{6L} \\
&= r_a(\mathbf{z}) + \frac{1}{3} r_a(\mathbf{z}) \\
&= \frac{4}{3} r_a(\mathbf{z}),
\end{aligned} \tag{158}
$$

thus $\|z - u\| \leq 3r_a(\mathbf{u})/4$ implies $\|u - z\| \leq r_a(\mathbf{z})$. Therefore (a) holds.

For (b) in (156), note that for $\|z - u\| \leq 3r_a(\mathbf{u})/4$, using Assumption 2(d) again,

$$
\eta^*(\mathbf{z}) - \eta_a(\mathbf{z}) \leq \eta^*(\mathbf{u}) - \eta_a(\mathbf{u}) + L\frac{3r_a(\mathbf{u})}{4} + L\frac{3r_a(\mathbf{u})}{4} = \eta^*(\mathbf{u}) - \eta_a(\mathbf{u}) + \frac{3}{2}L r_a(\mathbf{u}). \tag{159}
$$

Then

$$
\begin{aligned}
\frac{g(\mathbf{z})}{g(\mathbf{u})} &= \frac{[(\eta^*(\mathbf{u}) - \eta_a(\mathbf{u})) \vee \epsilon]^d}{[(\eta^*(\mathbf{z}) - \eta_a(\mathbf{z})) \vee \epsilon]^d} \\
&\geq \frac{[(\eta^*(\mathbf{u}) - \eta_a(\mathbf{u})) \vee \epsilon]^d}{[(\eta^*(\mathbf{u}) - \eta_a(\mathbf{u}) + \frac{3}{2}L r_a(\mathbf{u})) \vee \epsilon]^d} \\
&\geq \left(\frac{4}{5}\right)^d, 
\end{aligned} \tag{160}
$$

in which the last step comes from the definition of $r_a$ in (77).

(c) uses (77) and the definition of $g$ in (81).

For (d), recall the statement of Lemma 7, $\epsilon = 4b$. Therefore, if $\eta^*(\mathbf{u}) - \eta_a(\mathbf{u}) > \epsilon$, then $\eta^*(\mathbf{u}) - \eta_a(\mathbf{u}) - 2b > (\eta^*(\mathbf{u}) - \eta_a(\mathbf{u}))/2$.

### I.5. Proof of Lemma 8

$$
\mathbb{E}\left[\int_{B(\mathbf{Z}, r_a(\mathbf{Z}))} q_a(\mathbf{u})(\eta^*(\mathbf{u}) - \eta_a(\mathbf{u}))du\right]
$$

$$
\overset{(a)}{\leq} \frac{4}{3}\mathbb{E}\left[\int_{B(\mathbf{Z}, r_a(\mathbf{Z}))} q_a(\mathbf{u})(\eta^*(\mathbf{z}) - \eta_a(\mathbf{z}))du\right]
$$

$$
\overset{(b)}{\leq} \frac{4}{3}(k+1)\mathbb{E}\left[(\eta^*(\mathbf{Z}) - \eta_a(\mathbf{Z}))\right]
$$

$$
= \frac{4}{3}(k+1)\int_{\mathcal{X}}(\eta^*(\mathbf{z}) - \eta_a(\mathbf{z}))g(\mathbf{z})dz
$$

$$
= \frac{4(k+1)}{3M_Z}\int_{\mathcal{X}}(\eta^*(\mathbf{z}) - \eta_a(\mathbf{z}))\frac{1}{[(\eta^*(\mathbf{z}) - \eta_a(\mathbf{z})) \vee \epsilon]^d}dz
$$

$$
= \frac{4(k+1)}{3M_Z}\left[\int_{\mathcal{X}}(\eta^*(\mathbf{z}) - \eta_a(\mathbf{z}))^{-(d-1)}\mathbf{1}(\eta^*(\mathbf{z}) - \eta_a(\mathbf{z}) > \epsilon)dz \right.
$$

$$
\left. + \frac{1}{\epsilon^d}\int_{\mathcal{X}}(\eta^*(\mathbf{z}) - \eta_a(\mathbf{z}))\mathbf{1}(\eta^*(\mathbf{z}) - \eta_a(\mathbf{z}) < \epsilon)dz\right]. \tag{161}
$$

For (a), from Assumption 2(d),

$$
\begin{aligned}
\eta^*(\mathbf{u}) - \eta_a(\mathbf{u}) &\leq \eta^*(\mathbf{z}) - \eta_a(\mathbf{z}) + 2Lr_a(\mathbf{z}) \\
&= \eta^*(\mathbf{z}) - \eta_a(\mathbf{z}) + 2L\frac{\eta^*(\mathbf{z}) - \eta_a(\mathbf{z}) - 2b}{6L} \\
&\leq \frac{4}{3}(\eta^*(\mathbf{z}) - \eta_a(\mathbf{z})).
\end{aligned} \tag{162}
$$

(b) comes from (80).

The first term in the bracket in (161) can be bounded by

$$
\int_{\mathcal{X}}(\eta^*(\mathbf{z}) - \eta_a(\mathbf{z}))^{-(d-1)}\mathbf{1}(\eta^*(\mathbf{z}) - \eta_a(\mathbf{z}) > \epsilon)dz
$$

$$
\overset{(a)}{\leq} \frac{1}{c}\int_{\mathcal{X}}(\eta^*(\mathbf{z}) - \eta_a(\mathbf{z}))^{-(d-1)}\mathbf{1}(\eta^*(\mathbf{z}) - \eta_a(\mathbf{z}) > \epsilon)f(\mathbf{z})dz
$$

$$
\overset{(b)}{=} \frac{1}{c}\mathbb{E}\left[(\eta^*(\mathbf{X}) - \eta_a(\mathbf{X}))^{-(d-1)}\mathbf{1}(\eta^*(\mathbf{X}) - \eta_a(\mathbf{X}) > \epsilon)\right]
$$

$$
= \frac{1}{c}\int_0^\infty \mathrm{P}(\epsilon < \eta^*(\mathbf{X}) - \eta_a(\mathbf{X}) < t^{-\frac{1}{d-1}})dt
$$

$$
\leq \frac{1}{c}\int_0^{\epsilon^{-(d-1)}} \mathrm{P}(\eta^*(\mathbf{X}) - \eta_a(\mathbf{X}) < t^{-\frac{1}{d-1}})dt. \tag{163}
$$

(a) comes from Assumption 2, which requires that $f(\mathbf{x}) \geq c$ all over the support. In (b), the random variable $X$ follows a distribution with pdf $f$.

If $d > \alpha + 1$, then from Assumption 2(b),

$$
(163) \leq \frac{C_\alpha}{c}\int_0^{\epsilon^{-(d-1)}} t^{-\frac{\alpha}{d-1}}dt = \frac{C_\alpha(d-1)}{c(d-1-\alpha)}\epsilon^{\alpha+1-d}. \tag{164}
$$

If $d = \alpha + 1$, then

$$(163) \leq \frac{1}{c} \int_0^1 dt + \frac{C_\alpha}{c} \int_1^{\epsilon^{-(d-1)}} t^{-\frac{\alpha}{d-1}} dt = \frac{1}{c} + \frac{C_\alpha(d-1)}{c} \ln \frac{1}{\epsilon}. \tag{165}$$

If $d < \alpha + 1$, then

$$(163) \leq \frac{1}{c} \int_0^1 dt + \frac{C_\alpha}{c} \int_1^{\epsilon^{-(d-1)}} t^{-\frac{\alpha}{d-1}} dt \leq \frac{1}{c} + \frac{C_\alpha(d-1)}{c(\alpha+1-d)}. \tag{166}$$

Now it remains to bound the second term in (161):

$$
\begin{aligned}
\int_{\mathcal{X}} (\eta^*(\mathbf{z}) - \eta_a(\mathbf{z}))\mathbf{1}(\eta^*(\mathbf{z}) - \eta_a(\mathbf{z}) < \epsilon)dz &\leq \frac{1}{c}\mathbb{E}[(\eta^*(\mathbf{X}) - \eta_a(\mathbf{X}))\mathbf{1}(\eta^*(\mathbf{X}) - \eta_a(\mathbf{X}) < \epsilon)] \\
&\leq \frac{C_\alpha}{c}\epsilon^{\alpha+1}.
\end{aligned}
\tag{167}
$$

Therefore, from (161), (164), (165), (166) and (167),

$$\mathbb{E}\left[\int_{B(\mathbf{Z}, r_a(\mathbf{Z}))} q_a(\mathbf{u})(\eta^*(\mathbf{u}) - \eta_a(\mathbf{u}))du\right] \lesssim \begin{cases} \frac{k}{M_Z c}\epsilon^{\alpha+1-d} & \text{if} \quad d > \alpha + 1 \\ \frac{k}{M_Z c} \ln \frac{1}{\epsilon} & \text{if} \quad d = \alpha + 1 \\ \frac{k}{M_Z c} & \text{if} \quad d < \alpha + 1. \end{cases} \tag{168}$$

### I.6. Proof of Lemma 12

We prove Lemma 12 by contradiction. Suppose now that $n(x, a, r_a(\mathbf{x})) > n_a(\mathbf{x})$. Let $t$ be the last sample falling in $B(x, r_a(\mathbf{x}))$, i.e.

$$t := \max\{j | \|\mathbf{X}_j - \mathbf{x}\| \leq r_a(\mathbf{x}), A_j = a\}. \tag{169}$$

We first show that $\rho_{a,t}(\mathbf{x}) \leq 2r_a(\mathbf{x})$. From (169) and the condition $n(x, a, r_a(\mathbf{x})) > n_a(\mathbf{x})$, before time step $t$, there are already at least $n_a(\mathbf{x})$ steps in $B(x, r_a(\mathbf{x}))$. Note that $\|\mathbf{X}_t - \mathbf{x}\| \leq r_a(\mathbf{x})$, thus $B(x, r_a(\mathbf{x})) \subseteq B(\mathbf{X}_t, 2r_a(\mathbf{x}))$. Therefore, there are already at least $n_a(\mathbf{x})$ samples with action $a$ in $B(\mathbf{X}_t, 2r_a(\mathbf{x}))$. Recall that $\rho_{a,t}(\mathbf{x}) = \rho_{a,t,k_{a,t}(\mathbf{x})}(\mathbf{x})$. If $\rho_{a,t}(\mathbf{x}) > 2r_a(\mathbf{x})$, then $k_{a,t}(\mathbf{x}) > n_a(\mathbf{x})$. From (16),

$$L\rho_{a,t}(\mathbf{x}) = L\rho_{a,t,k_{a,t}(\mathbf{x})}(\mathbf{x}) \leq \sqrt{\frac{\ln T}{k_{a,t}(\mathbf{x})}} \leq \sqrt{\frac{\ln T}{n_a(\mathbf{x})}} = 2Lr_a(\mathbf{x}), \tag{170}$$

then contradiction occurs. Therefore $\rho_{a,t}(\mathbf{x}) \leq 2r_a(\mathbf{x})$.

From Lemma 11, under $E$,

$$\hat{\eta}_{a,t}(\mathbf{x}) \leq \eta_a(\mathbf{x}) + 2\sqrt{\frac{2\sigma^2}{n_a(\mathbf{x})} \ln(dT^{2d+3}|\mathcal{A}|)} + 2Lr_a(\mathbf{x}). \tag{171}$$

Since action $a$ is selected at time $t$, from Lemma 11,

$$\hat{\eta}_{a,t}(\mathbf{x}) \geq \hat{\eta}_{a^*(\mathbf{x}),t}(\mathbf{x}) \geq \eta^*(\mathbf{x}). \tag{172}$$

Combining (171) and (172) yields

$$2\sqrt{\frac{2\sigma^2}{n_a(\mathbf{x})} \ln(dT^{2d+3}|\mathcal{A}|)} + 2Lr_a(\mathbf{x}) \geq \eta^*(\mathbf{x}) - \eta_a(\mathbf{x}). \tag{173}$$

We now derive an inequality that contradicts with (173). From (112) and (113),

$$
\begin{aligned}
2\sqrt{\frac{2\sigma^2}{n_a(\mathbf{x})} \ln(dT^{2d+3}|\mathcal{A}|)} &= 2\sqrt{\frac{2\sigma^2}{C_1 \ln T} \ln(dT^{2d+3}|\mathcal{A}|)}(\eta^*(\mathbf{x}) - \eta_a(\mathbf{x})) \\
&\leq \frac{1}{2}\sqrt{\frac{\ln(dT^{2d+3}|\mathcal{A}|)}{(2d+3+\ln(d|\mathcal{A}|))\ln T}}(\eta^*(\mathbf{x}) - \eta_a(\mathbf{x})) \\
&< \frac{1}{2}(\eta^*(\mathbf{x}) - \eta_a(\mathbf{x})).
\end{aligned}
\tag{174}
$$

From (111),

$$2Lr_a(\mathbf{x}) = \frac{1}{\sqrt{C_1}}(\eta^*(\mathbf{x}) - \eta_a(\mathbf{x})) \leq \frac{1}{2}(\eta^*(\mathbf{x}) - \eta_a(\mathbf{x})). \tag{175}$$

From (174) and (175),

$$2\sqrt{\frac{2\sigma^2}{n_a(\mathbf{x})}\ln(dT^{2d+3}|\mathcal{A}|)} + 2Lr_a(\mathbf{x}) < \eta^*(\mathbf{x}) - \eta_a(\mathbf{x}). \tag{176}$$

(176) contradicts (173). Hence

$$n(x, a, r_a(\mathbf{x})) \leq n_a(\mathbf{x}). \tag{177}$$

## I.7. Proof of Lemma 13

$$
\begin{aligned}
&\mathbb{E}\left[\int_{B(\mathbf{Z},r_a(\mathbf{Z}))} q_a(\mathbf{u})(\eta^*(\mathbf{u}) - \eta_a(\mathbf{u}))du\right] \\
&\overset{(a)}{\geq} \int_{\mathcal{X}}\int_{B(u,2r_a(\mathbf{u})/3)} g(\mathbf{z})q_a(\mathbf{u})(\eta^*(\mathbf{u}) - \eta_a(\mathbf{u}))dzdu \\
&\geq \int_{\mathcal{X}}\left(\inf_{\|z-u\|\leq 2r_a(\mathbf{u})/3} g(\mathbf{z})\right)\left(\frac{2}{3}\right)^d r_a^d(\mathbf{u})q_a(\mathbf{u})(\eta^*(\mathbf{u}) - \eta_a(\mathbf{u}))du \\
&\overset{(b)}{\geq} \left(\frac{2}{3}\right)^d\left(\frac{3}{4}\right)^d \int_{\mathcal{X}} g(\mathbf{u})r_a^d(\mathbf{u})q_a(\mathbf{u})(\eta^*(\mathbf{u}) - \eta_a(\mathbf{u}))du \\
&= \frac{1}{2^d M_Z}\int_{\mathcal{X}} \frac{1}{[(\eta^*(\mathbf{u}) - \eta_a(\mathbf{u})) \vee \epsilon^d]} r_a^d(\mathbf{u})q_a(\mathbf{u})(\eta^*(\mathbf{u}) - \eta_a(\mathbf{u}))du \\
&\leq \frac{1}{2^d M_Z}\int_{\mathcal{X}} \mathbf{1}(\eta^*(\mathbf{u}) - \eta_a(\mathbf{u}) > \epsilon)\frac{1}{(\eta^*(\mathbf{u}) - \eta_a(\mathbf{u}))^d}\frac{(\eta^*(\mathbf{u}) - \eta_a(\mathbf{u}))^d}{(4L)^d} q_a(\mathbf{u})(\eta^*(\mathbf{u}) - \eta_a(\mathbf{u}))du \\
&\leq \frac{1}{2^{3d}L^d M_Z}\int_{\mathcal{X}} q_a(\mathbf{u})(\eta^*(\mathbf{u}) - \eta_a(\mathbf{u}))\mathbf{1}(\eta^*(\mathbf{u}) - \eta_a(\mathbf{u}) > \epsilon)du. \tag{178}
\end{aligned}
$$

For (a), if $\|u - z\| \leq r_a(\mathbf{z})$, then from the definition of $r_a$ in (111),

$$\frac{r_a(\mathbf{u})}{r_a(\mathbf{z})} = \frac{\eta^*(\mathbf{u}) - \eta_a(\mathbf{u})}{\eta^*(\mathbf{z}) - \eta_a(\mathbf{z})} \leq \frac{\eta^*(\mathbf{z}) - \eta_a(\mathbf{z}) + 2Lr_a(\mathbf{z})}{\eta^*(\mathbf{z}) - \eta_a(\mathbf{z})} = 1 + \frac{1}{\sqrt{C_1}} \leq \frac{3}{2}. \tag{179}$$

For (b),

$$
\begin{aligned}
\frac{g(\mathbf{z})}{g(\mathbf{u})} &= \frac{[(\eta^*(\mathbf{u}) - \eta_a(\mathbf{u})) \vee \epsilon]^d}{[(\eta^*(\mathbf{z}) - \eta_a(\mathbf{z})) \vee \epsilon]^d} \\
&\geq \frac{[(\eta^*(\mathbf{u}) - \eta_a(\mathbf{u})) \vee \epsilon]^d}{[(\eta^*(\mathbf{u}) - \eta_a(\mathbf{u}) + \frac{4}{3}Lr_a(\mathbf{u})) \vee \epsilon]^d} \\
&\geq \left(\frac{3}{4}\right)^d. \tag{180}
\end{aligned}
$$

## I.8. Proof of Lemma 14

$$\mathbb{E}\left[\int_{B(\mathbf{Z},r_a(\mathbf{Z}))} q_a(\mathbf{u})(\eta^*(\mathbf{u}) - \eta_a(\mathbf{u}))du\right]$$

$$\overset{(a)}{\leq} \frac{3}{2}\mathbb{E}\left[\int_{B(\mathbf{Z},r_a(\mathbf{Z}))} q_a(\mathbf{u})(\eta^*(\mathbf{z}) - \eta_a(\mathbf{z}))du\right]$$

$$\leq \frac{3}{2}\mathbb{E}\left[((n_a(\mathbf{Z})+1) \wedge (Tf(\mathbf{Z})r_a^d(\mathbf{Z})))(\eta^*(\mathbf{Z}) - \eta_a(\mathbf{Z}))\right]$$

$$= \frac{3}{2}\int ((n_a(\mathbf{z})+1) \wedge (Tf(\mathbf{z})r_a^d(\mathbf{z})))(\eta^*(\mathbf{z}) - \eta_a(\mathbf{z}))\frac{1}{M_Z[(\eta^*(\mathbf{z}) - \eta_a(\mathbf{z})) \vee \epsilon]^d}dz$$

$$= \frac{3}{2M_g}\left[\int \left(\frac{C_1 \ln T}{\eta^*(\mathbf{z}) - \eta_a(\mathbf{z})} + \eta^*(\mathbf{z}) - \eta_a(\mathbf{z})\right)\frac{1}{(\eta^*(\mathbf{z}) - \eta_a(\mathbf{z}))^d}\mathbf{1}(\eta^*(\mathbf{z}) - \eta_a(\mathbf{z}) > \epsilon)dz\right.$$

$$\left. + \int Tf(\mathbf{z})r_a^d(\mathbf{z})(\eta^*(\mathbf{z}) - \eta_a(\mathbf{z}))\frac{1}{\epsilon^d}\mathbf{1}(\eta^*(\mathbf{z}) - \eta_a(\mathbf{z}) \leq \epsilon)dz\right]$$

$$\lesssim \frac{1}{M_Z}\mathbb{E}\left[(\eta^*(\mathbf{Z}) - \eta_a(\mathbf{Z}))^{-(d+1)}\mathbf{1}(\eta^*(\mathbf{Z}) - \eta_a(\mathbf{Z}) > \epsilon)\right]\ln T$$

$$+ \frac{T}{\epsilon^d}\mathbb{E}\left[(\eta^*(\mathbf{Z}) - \eta_a(\mathbf{Z}))^{d+1}\mathbf{1}(\eta^*(\mathbf{Z}) - \eta_a(\mathbf{Z}) \leq \epsilon)\right]$$

$$\lesssim \frac{1}{M_Z}\left(\epsilon^{\alpha-d-1}\ln T + T\epsilon^{1+\alpha}\right). \tag{181}$$

For (a),

$$\eta^*(\mathbf{u}) - \eta_a(\mathbf{u}) \leq \eta^*(\mathbf{z}) - \eta_a(\mathbf{z}) + 2Lr_a(\mathbf{z})$$

$$\leq \eta^*(\mathbf{z}) - \eta_a(\mathbf{z}) + \frac{1}{\sqrt{C_1}}(\eta^*(\mathbf{z}) - \eta_a(\mathbf{z}))$$

$$\leq \frac{3}{2}(\eta^*(\mathbf{z}) - \eta_a(\mathbf{z})). \tag{182}$$

## I.9. Proof of Lemma 9

$$\mathbb{E}[f^{-p}(\mathbf{X})\mathbf{1}(a \leq f(\mathbf{X}) < b)] = \int_0^\infty \mathrm{P}(f(\mathbf{X}) < t^{-\frac{1}{p}}, a \leq f(\mathbf{X}) < b)dt$$

$$= \int_0^{b^{-p}} \mathrm{P}(f(\mathbf{X}) < b)dt + \int_{b^{-p}}^{a^{-p}} \mathrm{P}(f(\mathbf{X}) < t^{-\frac{1}{p}})dt$$

$$\leq C_\beta b^{\beta-p} + C_\beta \int_{b^{-p}}^{a^{-p}} t^{-\frac{\beta}{p}}dt. \tag{183}$$

If $p > \beta$, i.e. $\beta/p < 1$, then

$$(183) \leq C_\beta b^{\beta-p} + \frac{C_\beta}{1 - \beta/p}(a^{-p})^{1-\frac{\beta}{p}} \sim a^{\beta-p}. \tag{184}$$

If $p < \beta$, then

$$(183) \leq C_\beta b^{\beta-p} + C_\beta \int_{b^{-p}}^\infty t^{-\frac{\beta}{p}}dt = C_\beta b^{\beta-p} + \frac{C_\beta}{\beta/p - 1}b^{\beta-p} \sim b^{\beta-p}. \tag{185}$$

If $p = \beta$, then

$$(183) \leq C_\beta b^{\beta-p} + C_\beta \ln \frac{a^{-p}}{b^{-p}} \sim \ln \frac{b}{a}. \tag{186}$$

### I.10. Proof of Lemma 3

Our proof follows the proof of Lemma 3.1 in (Rigollet & Zeevi, 2010).

$$U = \sum_{t=1}^{T} (\eta^*(\mathbf{X}_t) - \eta_{A_t}(\mathbf{X}_t)), \tag{187}$$

and

$$V = \sum_{t=1}^{T} \mathbf{1}(\eta_{A_t}(\mathbf{X}_t) < \eta^*(\mathbf{X}_t)). \tag{188}$$

Then $R = \mathbb{E}[U]$ and $S = \mathbb{E}[V]$. For any $\delta > 0$,

$$
\begin{aligned}
U & \geq \delta \sum_{t=1}^{T} \mathbf{1}(\eta_{A_t}(\mathbf{X}_t) < \eta^*(\mathbf{X}_t)) \mathbf{1}(|\eta^*(\mathbf{X}_t) - \eta_{A_t}(\mathbf{X}_t)| > \delta) \\
& \geq \delta \left[ V - \sum_{t=1}^{T} \mathbf{1} \left( A_t \neq a^*(\mathbf{X}_t), |\eta^*(\mathbf{X}_t) - \eta_{A_t}(\mathbf{X}_t)| \leq \delta \right) \right].
\end{aligned}
\tag{189}
$$

Take expectations, we have

$$R \geq \delta S - T C_\alpha \delta^{\alpha+1}. \tag{190}$$

Now we minimize the right hand side of (190). By making the derivative to be zero, let

$$\delta = \left( \frac{S}{(\alpha+1)T C_\alpha} \right)^{\frac{1}{\alpha}}. \tag{191}$$

Then

$$
\begin{aligned}
R & \geq S \left( \frac{S}{(\alpha+1)T C_\alpha} \right)^{\frac{1}{\alpha}} - T C_\alpha \left( \frac{S}{(\alpha+1)T C_\alpha} \right)^{\frac{\alpha+1}{\alpha}} \\
& = \frac{\alpha S}{\alpha+1} \left( \frac{S}{(\alpha+1)T C_\alpha} \right)^{\frac{1}{\alpha}} \\
& = C_0 S^{\frac{\alpha+1}{\alpha}} T^{-\frac{1}{\alpha}}.
\end{aligned}
\tag{192}
$$

The proof is complete.

