# OpenReview forum: "Contextual Bandits for Unbounded Context Distributions"
_ICML.cc/2025/Conference — ICML 2025 poster_

### Official Review · Reviewer_DUvP · 2025-03-03

**Overall Recommendation:** 3

**Summary:**

Stochastic contextual bandits, where there is a set of K actions, and at each round $t$ the learner observes the current context $X_t$ generated from the fixed distribution. This paper considers the nonparametric setting with the standard assumption of zero-mean noise and Lipsthiz reward function. The authors investigate the min-max lower bound for unbounded supports by extending the result of (Rigollet & Zeevi, 2010).
Then, as a simple method, they proposed the k-NN UCB with fixed k and analyzed both cases of bounded and unbounded support.
They further devised the k-NN UCB with adaptive k to improve the regret bound matching the lower bound up to log factors. Experimental evidence was provided using the synthetic setting and MNIST dataset.

**Claims And Evidence:**

Strength
- Theorem 3 ( k-NN UCB with fixed k for bounded support)
recovers the result of (Guan & Jiang, 2018) when $\alpha=0$  and improve the result when $\alpha>0$.
- Theorem  5 ( k-NN UCB with adaptive  k for bounded support) is nearly optimal in both regimes with $d > \alpha+1$ or $d \leq \alpha+1$
- Theorem  4 ( k-NN UCB with fixed k for unbounded support) and Theorem  6 ( k-NN UCB with adaptive k for unbounded support) are the novel results for unbounded support.

**Essential References Not Discussed:**

The following paper could be cited in the revised version.


SUK, J. and KPOTUFE, S. (2021). Self-Tuning Bandits over Unknown Covariate-Shifts. In Proceedings of the 32nd International Conference on Algorithmic Learning Theory

**Experimental Designs Or Analyses:**

Experimental evidence was provided using the synthetic setting with two actions and subgaussian context distribution and real-world dataset where MNIST figure corresponds to the context and there are 10 actions.

**Methods And Evaluation Criteria:**

k-NN UCB is a simple and general method to deal with contextual bandits in non-parametric case. The main proposed method with adaptive choice of k is still simple but it is reasonable to deal with variance-bias trade-off adaptively.

**Other Comments Or Suggestions:**

See Other Strengths And Weaknesses

**Other Strengths And Weaknesses:**

Writing Quality:

Although critical issues have not been found, the writing quality is not the best. Some definitions are missing, and there are typos.

- (just a comment): p.1 right 053: Instead of “simple method”, you could specify that the method is a $k$-NN.
- Assumption 1 (a), you need “for all $X \in \mathcal{X}$”
- Assumption 1 (b), you need “for all $\lambda  \in \mathbb{R}$”
- Assumption 1 (b), what is $W_i$? It should be $W_t$.
- p.3. line 135, you are mixing $t$ and $u$; $t$ is undefined, I guess you want to use $\forall u>0$ here.
- line 158 right: $c$ is undefined.
- line 296 right: $x$ should be bold.
- Ling 259 right: $T_a(t-1)$ is undefined. I guess you mean $n_a(t-1)$.
- In (16): $k_t(x)$ depends on action $a$. It should be $k_{a,t}(x)$.
- In Remark 2: $N$ is undefined.
- In (92): you have a typo.
- In Lemmas 5,6, and 12: You need to specify or refer to the definition of event $E$.

**Questions For Authors:**

1. Is Assumption 3(b) common in the literature for unbounded support in contextual bandits?
When the suboptimal gap $\Delta_\min$ or $\Delta_a$ is large, the problem becomes easier to identify the optimal action as the regret bound usually depends on $1/\Delta$. But it seems that we need the opposite condition.


2. The reviewer is not familiar with non-parametric bandits, but I found that the proof techniques are very simple and standard as used in UCB algorithms. What is the main technical challenge compared with standard linear contextual bandits? Since here the Lipshitz reward function is only considered, the proof technique is essentially similar to the case of the linear reward function. When we deal with heavy-tailed distribution, which lemmas are novel and crucial in the analysis of non-parametric bandits?

**Relation To Broader Scientific Literature:**

The heavy-tailed context distribution in nonparametric contextual bandits was discussed in the paper.

**Theoretical Claims:**

The following are minor concerns, and I appreciate any feedback in rebuttal:

- More discussion of Theorem 4, e.g., comparison with Theorem 3 could be added in the revised version.
- The lower bound analysis does not capture the dependence of $|\mathcal{A}|$, although existing work only discusses the case of two actions. The regret bounds depend on the linear dependence of $|\mathcal{A}|$, which encourages us to investigate the optimality of the number of actions.
- (Just a comment) Proof Sketch of Theorem 2 is a higher-level idea and could be more detailed.

---

> ### Author Rebuttal · Authors · 2025-03-31
>
> Thanks the reviewer for your careful reading of this paper!
>
> We respond to your comments as follows.
>
> 1. We have read the paper you have mentioned: Self-Tuning Bandits over Unknown Covariate-Shifts. In ICALT. We think that this paper is indeed highly relevant, so we will compare it with our paper in our revised version.
>
> 2. Assumption 1(a): here $X$ is a random variable, instead of a fixed value. The randomness in probability $P(0<\eta^*(X)-\eta_a(X)<u)$ also comes from the randomness in X,  Therefore, we do not need "for all x\in \mathcal{X}".
>
> 3. Assumption 1(b)：Thanks for these two comments. $W_i$ should be $W_t$, and $\lambda \in \mathbb{R}$ needs to be mentioned.
>
> 4. Line 158: $c$ is some constant. We will clarify it.
>
> 5. Line 296 right: Yes, $x$ should be bold
>
> 6. Line 259 right: This notation comes from Guan et al. 2018. We will clarify the meaning of this notation.
>
> 7. Remark 2: $N$ is the number of samples for nonparametric classification with i.i.d samples. We will clarify it.
>
> 8. (92): Yes, it should be $T^{2d+2}$, instead of $T(2d+2)$.
>
> 9. Lemma 5,6 and 12: Thanks. We will clarify them.
>
> **Question 1: About assumption 3(b)**
>
> This is not common in previous literatures, since existing works only discuss bounded context support. We think you have raised a very good question. It is indeed a bit counterintuitive that we need small suboptimal gap $\Delta_a$. For unbounded context support, there exists some region with very low density $f(\mathbf{x})$, such that even if the suboptimal gap is large, we still can not identify the optimal action. Therefore, noting that the regret is upper bounded by the suboptimal gap, instead of trying to identify the optimal action, now we give an upper bound of the suboptimal gap, so that the regret (with unidentified optimal action) can be controlled. This is an important distinction between bounded and unbounded context support. For bounded support, since the density is lower bounded (Assumption 2: $f(\mathbf{x})\geq c$), we hope that suboptimal gap is as large as possible. However, for unbounded case, things become more complex and large suboptimal gap does not always make the problem easier.
>
> **Question2: Novelty**
>
> The main novelty is the treatment of tails. The proof of theorem 2 is novel, as it requires treatment of heavy tails.
>
> In Appendix B, the "expected sample density" has not been proposed by existing analysis.
>
> Lemma 5,6,7,8 and Appendix E are entirely new.
>
> We guess that some lemmas (such as Lemma 4) appears to be similar to existing works, which leaves an impression that the proof is simple and standard. However, these lemmas are tools that are required for the completeness of analysis. For all other parts of our theoretical analysis, different techniques are used.
>
> The main technical challenge is to achieve both exploration-exploitation tradeoff and bias-variance tradeoff. In bounded context space, one only needs to achieve the former one.

---

> > ### Comment · Reviewer_DUvP · 2025-04-04
> >
> > Thank you for your feedback. I have taken a brief look at Reeve et al. (2018), and I believe that handling unbounded contexts requires novel analysis. At this point, I have no major concerns.

---

### Official Review · Reviewer_nDm2 · 2025-03-10

**Overall Recommendation:** 4

**Summary:**

Contextual bandit is important in recommendation systems, healthcare, etc. Existing works focus primarily on linear bandits (or other parametric bandits). While some papers study nonparametric bandits, they assume that the support is bounded. In this work, the authors study nonparametric contextual bandit with unbounded context support. The paper proposes two methods: fixed k and adaptive k. For the latter approach, k is adaptively selected to balance the bound of bias and variance. According to the theoretical analysis, the fixed k method achieves minimax optimal rate under some parameter regimes. The adaptive k method achieves minimax optimal rate for all regimes.

**Claims And Evidence:**

I think that the claims are clear. The authors provide sufficient theoretical analysis to validate their claims.

**Essential References Not Discussed:**

This paper has all essential references.

**Experimental Designs Or Analyses:**

The experiments in this paper mainly use synthesized data. I think that for evaluating nonparametric statistics, it is reasonable to use synthesized data, for the convenience of analysis. In general, I think that the the experiments are sound enough.

**Methods And Evaluation Criteria:**

The proposed methods and evaluation make sense for the problem at hand. This paper discusses nonparametric bandits, therefore knn method is natural. The problem is how to determine the UCB, as well as the selection of k. I think that the authors did a great job in figuring out a proper way, such that k is selected adaptively based on the bias and variance bounds. Based on my understanding, I think that the proposed method makes sense.

**Other Comments Or Suggestions:**

In general, I think that this paper is well written. However, I think that the proof needs to be polished, and more intuition need to be provided. Hope that authors can respond to my concerns raised above. I will consider improving my score if authors provide a good response.

**Other Strengths And Weaknesses:**

[Strengths]

1. Importance: This paper addresses an important problem. Existing analysis on contextual bandits focus on bounded support. However, unbounded supports are more common in practice.

2. Novelty: The adaptive method is novel. It achieves two tradeoffs simultaneously: exploration-exploitation tradeoff and bias-variance tradeoff.

[Weaknesses]

Some points in the proof are not clear to me. $C_\alpha$ is considered to be constant in eq.(50). However, In eq.(38), it seems that C_\alpha has a dependence $h^{d-\alpha}$.

Moreover, it would be interesting to consider the case if contexts lie in a manifold. While bandit algorithms can not overcome curse of dimensionality in general, if the contexts have low intrinsic dimensionality, then the regret may still be controlled even if the overall dimensionality is high. Hope that authors can provide some ideas.

**Questions For Authors:**

See weakness

**Relation To Broader Scientific Literature:**

There are some scientific literatures on nonparametric contextual bandits. Previous works focus primarily on bounded context supports. This paper solves the problem of unbounded context supports. I think that this paper is an important extension to previous works.

**Theoretical Claims:**

The proofs of all theoretical claims are shown in the Appendix. I have briefly reviewed the proofs. It seems that the proof is correct.

A concern is that I do not fully understand why Lemma 3 can not be used in the proof of lower bound for unbounded case. My intuition is that Lemma 3 is quite general. So I am not sure why Lemma 3 can not be directly used to get the lower bound of regret with unbounded support.

---

> ### Author Rebuttal · Authors · 2025-03-31
>
> Thank you very much for your positive feedback on the importance and novelty of this paper. We reply to questions as follows.
>
> 1. $C_\alpha$ is a constant. In eq.(38), $K$ has $h^{d-\alpha}$ dependence. Throughout the paper, $C_\alpha$ remains a constant.
>
> 2. Thanks for the suggestion. We think that **all our results hold for intrinsic dimension $d$**. This means that even though the overall dimensionality is significantly larger than $d$, the result in all theorems in the paper still hold.
>
> In our revised paper, we will polish the proof further and provide more necessary intuition and explanation.

---

> > ### Comment · Reviewer_nDm2 · 2025-04-02
> >
> > Thanks for the rebuttal, which addresses my concerns. After reviewing the supplementary material further, I now understand the proof of the lower bound. The paper is solid and novel, so I’ve decided to raise my score to 4.
> >
> > I’ve also considered other reviewers' comments and the authors' feedback, particularly Reeve et al.'s paper. The method in this paper is novel and distinct from Reeve et al.'s approach. I generally agree with reviewer c4Ni's feedback, but Reeve et al.'s method performance hasn’t been analyzed for unbounded contexts, leaving its optimality uncertain. I hope the authors can provide further insights on this.

---

> > > ### Author Response · Authors · 2025-04-04
> > >
> > > Thank you very much for your further reply, as well as the score increase! Moreover, thanks for acknowledging the novelty compared with Reeve et al.'s approach.
> > >
> > > Yes, we agree that Reeve et al's method has not been analyzed for unbounded contexts, thus we can not claim affirmatively that this method is slower than the minimax rate. Our intuition is that the selection of k in Reeve's paper is merely a minimization of regression function estimate, plus the UCB. It does not achieve good exploitation/exploration tradeoff and bias-variance tradeoff simultaneously. In the future, we plan to derive a lower bound of Reeve et al.'s method to further validate all our claims.
> > >
> > > We are very glad to provide further response if you have remaining questions. Thanks!

---

### Official Review · Reviewer_c4Ni · 2025-03-10

**Overall Recommendation:** 3

**Summary:**

In this paper, the authors study contextual bandit problems under the Tsybakov margin condition. They consider settings where the context distribution is either bounded or unbounded but heavy-tailed. Compared to the literature, they work under a weaker version of the Tsybakov margin condition, allowing them to establish a stronger lower bound. They propose an approach equipped with $k$-nearest neighbor methods and UCB exploration. By incorporating an adaptable $k$ in the nearest neighbor methods, they show that their algorithm achieves a regret upper bound that matches their strengthened lower bound up to logarithmic factors under both the bounded and heavy-tailed context assumptions. Finally, experiments on both synthesized and real data are presented to demonstrate the practical performance of their algorithm.

**Claims And Evidence:**

All the claims are clear and proved.

**Essential References Not Discussed:**

They have cited all relevant papers to my knowledge.

**Experimental Designs Or Analyses:**

They apply their algorithm to both synthesized and real data.

**Methods And Evaluation Criteria:**

They achieve minimax regret bounds.

**Other Comments Or Suggestions:**

See strengths and weaknesses

**Other Strengths And Weaknesses:**

The main weakness of the paper is that the algorithm itself may not be a novel contribution. Throughout the paper, the authors compare their work only with Guan & Jiang (2018) but do not compare it with Reeve et al. (2018). However, I do not see a clear difference between the method proposed in this paper and the approach taken by Reeve et al. (2018) in the bounded context setting. It seems to me that the authors merely restate how the parameter $k$ is chosen. Additionally, the generalization to the unbounded but heavy-tailed distribution appears to be a natural extension that incorporates the tail bounds from the previous analysis. Thus, I am somewhat concerned about the technical contribution of the paper. It would be helpful to highlight the technical challenges or obstacles involved in extending the approach to the unbounded setting.

**Questions For Authors:**

- Is there anything prevents the algorithm proposed by Reeve et al. (2018) from being applied to this setting of unbounded context?

**Relation To Broader Scientific Literature:**

They extended previous approach

**Theoretical Claims:**

All the claims are clear and proved.

---

> ### Author Rebuttal · Authors · 2025-03-25
>
> Thanks for your review. We are encouraged that you agree that our claims are all clear and proved.
>
> Regarding the novelty of algorithm. **We disagree that "the authors merely restate how the parameter $k$ is chosen"**. For our adaptive method, we select $k$ according to eq.(16). For Reeve et al. (2018), $k$ is selected in Algorithm 1, in particular, in step 2(b). The UCB is defined in the page before Algorithm 1, with $\phi(t)$ not fully determined. As can be observed from our paper and Reeve et al.'s paper, the selection rule is quite different: we select $k$ as the maximum one that satisfies $L\rho_{a,t,j}(x)\leq \sqrt{\ln T/j}$, while Reeve et al.'s paper select $k$ by minimizing a new defined UCB. The UCB calculations are also different: we calculate UCB in eq.(17) in our paper, which is also different from Reeve et al.'s paper, in the equation before Algorithm 1.
>
> However, we do think that the question (about anything that prevents the algorithm in Reeve et al. from being used in unbounded context) raised by the reviewer is very valuable. Actually, if we only consider implementation, without considering the theoretical bounds, then the algorithm can indeed be applied into unbounded contexts. However, the convergence rate is not analyzed right now, and we believe that it will be much more complex than our method. It is unknown whether the method proposed in Reeve et al. will match the lower bound. **Therefore, to the best of our knowledge, our work is still the first one to establish the minimax lower bound of contextual bandits with unbounded contexts, and provide algorithms with matching upper bounds.**
>
> In addition, although less important, we would like to mention that our selection of $k$ requires $O(\ln T)$ time, while Reeve et al's method requires $O(T)$ time. Therefore we also have advantages in time complexity.

---

### Official Review · Reviewer_syqz · 2025-03-16

**Overall Recommendation:** 3

**Summary:**

The paper studies the setting of contextual bandits for unbounded context set. The paper then proposed an idea of algorithm design based on k-nearest neighbor, with a special design of the optimism term. With an adaptive choice of $k$, the algorithm achieves the minimal-optimal regret.

**Claims And Evidence:**

All claims are well supported by evidence.

**Essential References Not Discussed:**

N/A

**Experimental Designs Or Analyses:**

I checked the experiment, which is well-designed.

**Methods And Evaluation Criteria:**

The proposed methods are solid.

**Other Comments Or Suggestions:**

In the discussion after Theorem 6, the case where $\beta$ goes to infinity reduces to Theorem 5 is interesting. It looks like the same relationship holds for Theorem 4 and Theorem 3.

**Other Strengths And Weaknesses:**

Strengths:
1. This paper is well written, with sufficient discussions on assumptions and main theorems.

Weaknesses:
1. My major concern is whether unbounded context distribution is an important objective to study. In our works about contextual bandits, the assumption of bounded **action space** is often made, but the theoretical results usually apply to the practical settings where the action space is unbounded. I think the setting of unbounded context set would be of special importance either if the bounded setting and the unbounded setting are fundamentally different by some obvious intuitions, or if the practical results of unbounded context set departs too much from the theory about bounded context set. I would raise my score if this question is well addressed by the authors.
2. It is questionable whether the size of the action set, which is the major gap between the lower bound and the upper bound in Theorem 6, can be treated as a constant.

**Questions For Authors:**

1. How can we interpret the phase transition around $\alpha=d+1$ in Theorem 3?

**Relation To Broader Scientific Literature:**

This work is closely related to the literature studying nonparametric contextual bandits and using nearest neighbors (especially Guan and Jiang 2018), which is discussed in great detail in Section 2. The idea of using an adaptive $k$ is a new idea.

**Theoretical Claims:**

I did not check the proof.

---

> ### Author Rebuttal · Authors · 2025-03-31
>
> Thank you very much for acknowledging the writing of this paper. We respond to questions and weaknesses as follows.
>
> **1. Importance of unbounded context distribution**
>
> The bounded action space can be easily generalized to unbounded action space (assuming continuous action space). Unbounded action space does not involve too much additional technical difficulties.
>
> However, generalizing bounded context distribution to unbounded one is significantly different. We strongly agree with your comment **" I think the setting of unbounded context set would be of special importance either if the bounded setting and the unbounded setting are fundamentally different by some obvious intuitions, or if the practical results of unbounded context set departs too much from the theory about bounded context set."** We think that these two reasons both hold.
>
> **(1) Fundamental difference by intuition.** In bounded context space, we only need to achieve a tradeoff between exploration and exploitation. However, with unbounded context space, since the sample density is crucially different in different regions, we need to also achieve a better tradeoff between bias and variance. As emphasized in the abstract and introduction, the challenge is to achieve both exploration-exploitation tradeoff and bias-variance tradeoff simultaneously. This problem does not exist for bounded context space.
>
> **(2) Difference in practical results.** We refer to Theorem 6. The bounded context case corresponds to Theorem 6 with $\beta \rightarrow \infty$, which yields a $O(T^{1-\frac{\alpha+1}{d+2}})$ regret. As long as $\beta$ is not infinite, the regret bound is clearly different from the bounded context case. In particular, with $\beta<1/(d+2)$, the difference is more significant as the main cause of regret comes from the tail, instead of exploration.
>
>
> **2. Whether the size of action set is treated as constant**
>
> It is common to treat size of action set in related works. See the following results as examples:
>
> Shah, Devavrat, and Qiaomin Xie. "Q-learning with nearest neighbors." Advances in Neural Information Processing Systems 31 (2018).
>
> Zhao, Puning, and Lifeng Lai. "Minimax optimal q learning with nearest neighbors." IEEE Transactions on Information Theory (2024).
>
> **3. Interpretation of phase transition in Theorem 3**
>
> As discussed in the response of weaknesses 1, in unbounded state space, we need to achieve both exploration-exploitation tradeoff and bias-variance tradeoff. Fixed $k$ method fails to achieve the latter one, as in heavy-tailed contexts, bias-variance tradeoff is more complicated and smaller $k$ may be necessary.
>
> **This result further explains the importance of studying unbounded context supports.** At first glance, one may think that it is simple to generalize fixed k method to unbounded context, but our analysis show that such method is no longer optimal. We need to find new method to achieve minimax optimal regret.

---

> > ### Comment · Reviewer_syqz · 2025-04-01
> >
> > I appreciate the authors for the rebuttal! The response, especially the answer to Weakness 1, clearly addresses my concerns. I have raised my score, and have no more questions

---

> > > ### Author Response · Authors · 2025-04-04
> > >
> > > Thank you very much for your reply as well as the score increase! We will further revise the paper according to your comments. We are also very glad to reply to any further questions.

---

### Decision · Program_Chairs · 2025-05-01

**Decision:**

Accept (poster)

**Comment:**

This paper studies the nonparametric bandit problem with a specific focus on unbounded contexts. The setting of unbounded contexts turns out to be fundamentally different from the bounded ones because 1) unbounded contexts involve an additional bias-variance tradeoff on top of the exploration-exploitation tradeoff; 2) the resulting regret bounds are very different and could be dominated by the tail strength of the context distribution. This paper establishes the tight upper (based on k-NN algorithms) and lower bounds (up to log factors) for this problem.

All reviewers unanimously appreciate this paper: it studies a clear problem, proposes a sound algorithm, and establishes matching upper and lower bounds. The main concern is on the additional algorithmic and analytical novelty compared to the existing work on bounded contexts. Based on my own reading and the authors' rebuttal, the reviewers think the merits outweigh the weaknesses, which I concur and recommend (weak) acceptance.